# Expansion-assisted selective plane illumination microscopy for nanoscale imaging of centimeter-scale tissues

**Adam Glaser[1]\*[†], Jayaram Chandrashekar[1]\*[†], Sonya Vasquez[1], Cameron Arshadi[1], Rajvi Javeri[1], Naveen Ouellette[1], Xiaoyun Jiang[1], Judith Baka[1], Gabor Kovacs[1], Micah Woodard[1], Shamishtaa Seshamani[1], Kevin Cao[1], Nathan Clack[2], Andrew Recknagel[1], Anna Grim[1], Pooja Balaram[3], Emily Turschak[3], Marcus Hooper[3], Alan Liddell[2], John Rohde[1], Ayana Hellevik[3], Kevin Takasaki[3], Lindsey Erion Barner[1], Molly Logsdon[1], Chris Chronopoulos[1], Saskia EJ de Vries[1], Jonathan T Ting[3], Steven Perlmutter[4], Brian E Kalmbach[3], Nikolai Dembrow[3], Bosiljka Tasic[3], R Clay Reid[3], David Feng[1], Karel Svoboda[1]**

[1]Allen Institute for Neural Dynamics, Seattle, United States; [2]Chan Zuckerberg Initiative, Redwood City, United States; [3]Allen Institute for Brain Science, Seattle, United States; [4]University of Washington, Seattle, United States

**\*For correspondence:**
adam.glaser@alleninstitute.org
(AG);
jayaramc@alleninstitute.org (JC)

[†]These authors contributed
equally to this work

**Competing interest:** See page
21

**Reviewing Editor:** Melike
Lakadamyali, University of
Pennsylvania, United States

## eLife Assessment

The ExA-SPIM methodology developed here and characterized and supported by **convincing** evidence is an **important** development for the field of light sheet microscopy as the new technology provides an impressive field of view making it possible to image the entire expanded mouse brain at cellular and subcellular resolution.

**Abstract** Recent advances in tissue processing, labeling, and fluorescence microscopy are providing unprecedented views of the structure of cells and tissues at sub-diffraction resolutions and near single molecule sensitivity, driving discoveries in diverse fields of biology, including neuroscience. Biological tissue is organized over scales of nanometers to centimeters. Harnessing molecular imaging across intact, three-dimensional samples on this scale requires new types of microscopes with larger fields of view and working distance, as well as higher throughput. We present a new expansion-assisted selective plane illumination microscope (ExA-SPIM) with aberration-free 1.5 µm×1.5 µm×3 µm optical resolution over a large field of view (10.6×8.0 mm$^2$) and working distance (35 mm) at speeds up to 946 megavoxels/s. Combined with new tissue clearing and expansion methods, the microscope allows imaging centimeter-scale samples with 375 nm lateral and 750 nm axial resolution (4× expansion), including entire mouse brains, with high contrast and without sectioning. We illustrate ExA-SPIM by reconstructing individual neurons across the mouse brain, imaging cortico-spinal neurons in the macaque motor cortex, and visualizing axons in human white matter.

## Introduction

Biological tissue is organized over scales of nanometers to centimeters. Understanding individual cells and multi-cellular organization requires probing the architecture of tissue over these spatial scales simultaneously. However, standard microscope objectives with high resolutions have limited working distances (<1 mm) and fields of view (<1 mm). Imaging large tissue volumes at high resolutions

**Figure 1.** Breaking the volumetric imaging barrier. (**a**) Current fluorescence microscopy approaches are bounded by a volumetric imaging barrier (thick pink line; inspired by *Daetwyler and Fiolka, 2023*). Resolution is limited by the diffraction limit. The accessible imaging volume is limited by specifications of life sciences microscope objectives. The former can be surpassed by using tissue expansion, and the latter can be overcome by using highly engineered lenses from the electronics metrology industry. (**b**) The etendue (**G**) of 90% of life sciences objectives (magenta) is <1 mm$^2$ (*Zhang and Gross, 2019a*), apart from several custom lenses (green). In contrast, lenses developed for electronics metrology can have G>10 mm$^2$. The lens used in the ExA-SPIM system provides a field of view of 16.8 mm$^2$ with NA = 0.305 (G=19.65 mm$^2$). This etendue is comparable to the custom RUSH objective (*Fan et al., 2019*), but with twice the working distance and correction for liquid media. The RUSH, Schmidt (*Voigt et al., 2024*), Kyocera (https://www.ksoc.co.jp/en/seihin/immersion-objective/immersion-objective.html), and large etendue curved focal plane (*Tang et al., 2024*) lenses that lie outside of their colored zone are highlighted.

The online version of this article includes the following figure supplement(s) for figure 1:

**Figure supplement 1.** A continuum of NA and expansion factor combinations can achieve a desired effective resolution.

therefore requires physical sectioning and extensive tiling. Physical sectioning distorts the imaged tissue, and focal planes have optical distortions at the edges of the field of view (*Zhang and Gross, 2019b*). These factors complicate stitching across sections and tile boundaries at sub-micrometer resolutions which in turn increases the complexity and cost of downstream image analysis.

Because light aberration and scattering limit high-resolution microscopy in tissue, including two-photon microscopy, to depths of hundreds of micrometers, sectioning and tiling have traditionally been viewed as necessary to image large specimens (*Portera-Cailliau et al., 2005*; *Tsai et al., 2009*; *Oh et al., 2014*; *Economo et al., 2016*). However, advances in histological methods, including clearing (*Richardson and Lichtman, 2015*; *Spalteholz, 1914*; *Dodt et al., 2007*; *Tsai et al., 2009*; *Hama et al., 2011*; *Becker et al., 2012*; *Ertürk et al., 2012*; *Chung and Deisseroth, 2013*; *Ke et al., 2013*; *Susaki et al., 2014*; *Tainaka et al., 2014*; *Yang et al., 2014*; *Renier et al., 2014*; *Hou et al., 2015*; *Costantini et al., 2015*; *Chi et al., 2018*) and expansion for microscopy (ExM; *Chen et al., 2015*; *Chen et al., 2016*; *Chozinski et al., 2016*; *Ku et al., 2016*), now promise diffraction-limited imaging deep in tissue. Avoiding sectioning and reducing tiling requires overcoming the 'volumetric' imaging barrier of microscopy (i.e. the maximum volume that may be imaged; *Figure 1*). Expansion up to 20×has been demonstrated (*Chang et al., 2017*), which provides access to molecular spatial scales (10's of nm), approaching those of cryo electron microscopy (a few nm; *Cheng et al., 2015*; *Fernandez-Leiro and Scheres, 2016*; *Nogales and Scheres, 2015*). These ExM methods produce large, fragile three-dimensional samples, further emphasizing the need for overcoming the volumetric imaging barrier.

We combine a new microscope and methods for tissue clearing and expansion, which we jointly refer to as Expansion-Assisted Selective Plane Illumination Microscopy (ExA-SPIM). Unlike expansion lattice light-sheet microscopy (*Gao et al., 2023*), which is limited to small tissue volumes << 1 mm$^3$, ExA-SPIM works with expansion of large centimeter-scale tissue volumes, such as the mouse brain. Leveraging optics and detectors developed for the electronics metrology industry, ExA-SPIM has a field of view ~100 × larger (13.3 mm diameter) and a working distance ~10 × larger (35 mm) compared to objectives typically used for biological microscopy, while retaining a relatively high numerical aperture (NA)=0.305. When combined with 3×expansion, the system achieves an effective resolution of ~0.5 µm laterally, and ~1 µm axially, at imaging speeds of up to 946 megavoxels/s. Imaging with diffraction-limited resolution throughout centimeter-scale specimens requires tissue

samples with small index of refraction variations (*Weiss et al., 2021*). Tailoring the expansion factor allows fine-tuning effective resolution and reduced light collection efficiency for specific tissue types and scientific questions (*Figure 1—figure supplement 1*).

We apply ExA-SPIM to imaging and reconstructing mammalian neurons. Axonal arbors of individual neurons are complex, branched structures that transmit electrical impulses over distances of centimeters, yet axons can be thinner than 100 nm (*Economo et al., 2016*; *Winnubst et al., 2019*). Axons contain numerous varicosities that make synapses with other neurons. Tracing the axonal arbors of single neurons is critical to define how signals are routed within the brain and is also necessary for classifying diverse neuron types into distinct types (*Economo et al., 2016*; *Wang et al., 2021*; *Winnubst et al., 2019*; *Xu et al., 2021*; *Zhang et al., 2021*). Large-scale projects based on single-cell transcriptomics in the mouse and human brain have revealed a great diversity of neuron types (*Hodge et al., 2019*; *Tasic et al., 2018*; *Yao et al., 2023*). However, the throughput of neuronal reconstructions has remained too low for single neuron reconstructions on a comparable scale, even in mice. Throughput is limited in part by speed and image quality of existing microscopy methods. We demonstrate that ExA-SPIM provides high-resolution fluorescence microscopy over teravoxel image volumes with minimal distortions, and thereby enables brain-wide imaging with high contrast, resolution, and speed.

## Results

We first describe the microscope optics and how they overcome specific requirements for multi-scale tissue imaging. We then outline a new histological method for clearing and expanding centimeter-scale specimens, although details are relegated to extensive protocols (*Ouellette et al., 2023*, dx.doi.org/10.17504/protocols.io.n92ldpwjxl5b/v1). Finally, we illustrate ExA-SPIM performance for imaging neurons in whole mouse brains and large samples of non-human primate and human cortex.

### ExA-SPIM microscope

The ideal fluorescence microscope for large-scale tissue imaging would provide: (1) nanoscale resolution, (2) over centimeter-scale volumes, (3) with minimal tiling and sectioning, (4) high isotropy in resolution and contrast, and (5) fast imaging speed. Multiple imaging systems have been developed within this space, each with unique advantages but inevitable trade-offs (*Huisken et al., 2004*; *Dodt et al., 2007*; *Wu et al., 2013*; *Kumar et al., 2014*; *Tomer et al., 2014*; *Economo et al., 2016*; *Narasimhan et al., 2017*; *Power and Huisken, 2017*; *Migliori et al., 2018*; *Chakraborty et al., 2019*; *Voigt et al., 2019*; *Voleti et al., 2019*; *Chen et al., 2020b*; *Chen et al., 2020a*; *Glaser et al., 2022*; *Wang et al., 2021*; *Xu et al., 2021*; *Zhang et al., 2021*; *Qi et al., 2023*; *Vladimirov et al., 2024*).

The choice of microscope objective is a critical design choice. Microscope objectives impose trade-offs between the smallest objects that can be resolved (i.e. the resolution), how much of the specimen can be observed at once (i.e. the field of view), and how thick of a specimen may be imaged (i.e. the working distance). These trade-offs are not based on physical law but reflect limitations of optical design, engineering, and lens manufacturing.

The trade-off between resolution and field of view is related to the etendue (G), which is proportional to the number of resolution elements of an optical system. The etendue is a quadratic function of the lens field of view (FOV) and numerical aperture (NA).

$$G = \frac{\pi}{4}(FOV \times NA)^2 \tag{1}$$

Similarly, for a given NA, the required lens diameter increases proportionally with the required working distance (WD). The ideal lens for large-scale volumetric microscopy would provide large G and WD values.

90% of commercially available objectives for biological microscopy have G<1 mm$^2$ (*Zhang and Gross, 2019a*), and the vast majority of commercially available lenses have lens diameters < 2 cm. Several attempts have been made to develop custom lenses for specific applications. The 'Mesolens' provides an etendue of G=6.25 mm$^2$, with FOV = 6 mm, NA = 0.47, and WD = 3 mm (*McConnell et al., 2016*), but difficulties with manufacturing have limited adoption (*McConnell, 2020*). An objective with an etendue of G=7.07 mm$^2$, FOV = 5 mm, NA = 0.6, and WD = 2.7 mm has been developed for two-photon microscopy in vivo (*Sofroniew et al., 2016*). However, this lens is customized for

infrared illumination and exhibits significant field curvature. Additional custom lenses have subsequently been developed for two-photon microscopy, although none of these lenses are suitable for large-scale fluorescence microscopy (*Ota et al., 2020*; *Rumyantsev et al., 2020*; *Yu et al., 2021*). A custom lens with a large etendue of G=18.86 mm$^2$, FOV = 14 mm, NA = 0.35, and WD = 19 mm has been developed for the real-time ultra-large-scale high-resolution (RUSH) microscope (*Fan et al., 2019*). Despite the large etendue, the lens is not commercially available and not designed for immersion, which is critical for imaging cleared or expanded tissues. Custom lenses such as the Schmidt (*Voigt et al., 2024*), Cousa (*Yu et al., 2024*), and Kyocera (https://www.ksoc.co.jp/en/seihin/immersion-objective/immersion-objective.html) lenses offer long working distances but small etendues. See *Appendix 1—table 1* for a summary of the existing custom microscopy objectives.

Rather than designing a highly customized lens from scratch, we leveraged engineering investments that have been made in a different domain. High-resolution, high-speed imaging is widely used within the machine vision and metrology industry, where optical microscopes are used to map defects in semiconductors and other electronic devices. As the physical size of electronic components (e.g. pixels on flat panel displays) has become smaller, lenses for this domain have been designed with increasing NA, accessing biologically relevant resolutions (<1 μm). For a given NA, these lenses have remarkably large fields of view (and thus high etendues), low-field curvature, minimal distortion, and chromatic correction throughout the visible wavelengths. A comparison of the etendue as a function of NA and working distance for these lenses, and commercial and custom life sciences lenses is summarized in *Figure 1*. A more detailed summary of many electronics metrology technologies is provided as *Supplementary file 1*. Despite their superb specifications, both the lenses and camera sensors are readily available, are manufactured in large quantities, and are cost effective. We investigated the performance of metrology optics and cameras for imaging biological tissues and adapted a particular set of machine vision optics for biological microscopy.

The detection path of the system uses a lens with 5.0×magnification, with diffraction-limited imaging at NA = 0.305, and a 16.8 mm field of view (G=19.65 mm$^2$) (VEO_JM DIAMOND 5.0×/F1.3, Schneider-Kreuznach, Germany) (*Figure 2a*). A 35-mm-thick glass beam splitter, normally used for co-axial illumination, is replaced with an optically equivalent thickness of liquid media. This enables spherical aberration-free imaging through 35–40 mm of liquid media (tunable for refractive indices between 1.33 and 1.56), including expanded hydrogels and cleared tissues. The lens is paired with a large-format CMOS sensor (VP-151MX, Vieworks Korea), based on the Sony IMX411 sensor, with a single-sided rolling shutter, 151 megapixels, 14192 (H)×10,640 (V), and a pitch of 3.76 μm. The camera captures a field of view of 10.6×8.0 mm (13.3 mm diagonal), capable of covering an entire 3×expanded mouse brain in only 15 tiles (*Figure 2b*). A conventional SPIM system would require 400+tiles to image an entire cleared brain at equivalent resolutions (*Figure 2—figure supplement 1*). A comparison of the selected lens and camera sensor with a state-of-the-art cleared tissue objective lens (Nikon 20×GLYC) and sCMOS camera (Hamamatsu Orca BT-Fusion) is shown in (*Figure 2—figure supplement 2*).

To improve axial resolution, our system synchronizes an axially-swept light sheet with the rolling shutter of the Sony IMX411 sensor (*Dean et al., 2015*). The CMOS sensor parallelizes readout across 14192 pixels per row, thereby achieving equivalent or greater pixel rates than typical scientific CMOS (sCMOS) sensors, even with an increased time per line and a relatively low frame rate (<6.4 Hz). As a result, the ExA-SPIM microscope operates at a higher imaging speed with comparable signal-to-noise ratio (SNR) to sCMOS-based systems (*Figure 2e*). See (*Figure 2—figure supplement 3*) for noise comparisons between the Sony IMX411 and sCMOS sensors. The low frame rate of the sensor facilitates accurate axially swept imaging using generic scanning hardware at an imaging speed of 946 megavoxels/sec. The excitation lens (1–290419, Navitar) is infinity-corrected (to enable axially swept excitation) and provides diffraction-limited resolution at NA = 0.133 over a 16 mm field of view. When fitted with a custom dipping cap, the excitation lens provides a working distance of 53 mm in water. The beam shaping in the excitation path is configured to deliver a light sheet with NA = 0.10 and width of 12.5 mm (full-width half-maximum). A full CAD layout of the ExA-SPIM system is shown in (*Figure 2—figure supplement 4*). Due to the large light-sheet, the ExA-SPIM system uses high excitation laser powers (1000+mW; *Figure 2—figure supplement 5*), resulting in light intensities typically used in SPIM microscopy. Photobleaching is not detrimental at these imaging conditions (*Figure 2—figure supplement 6*).

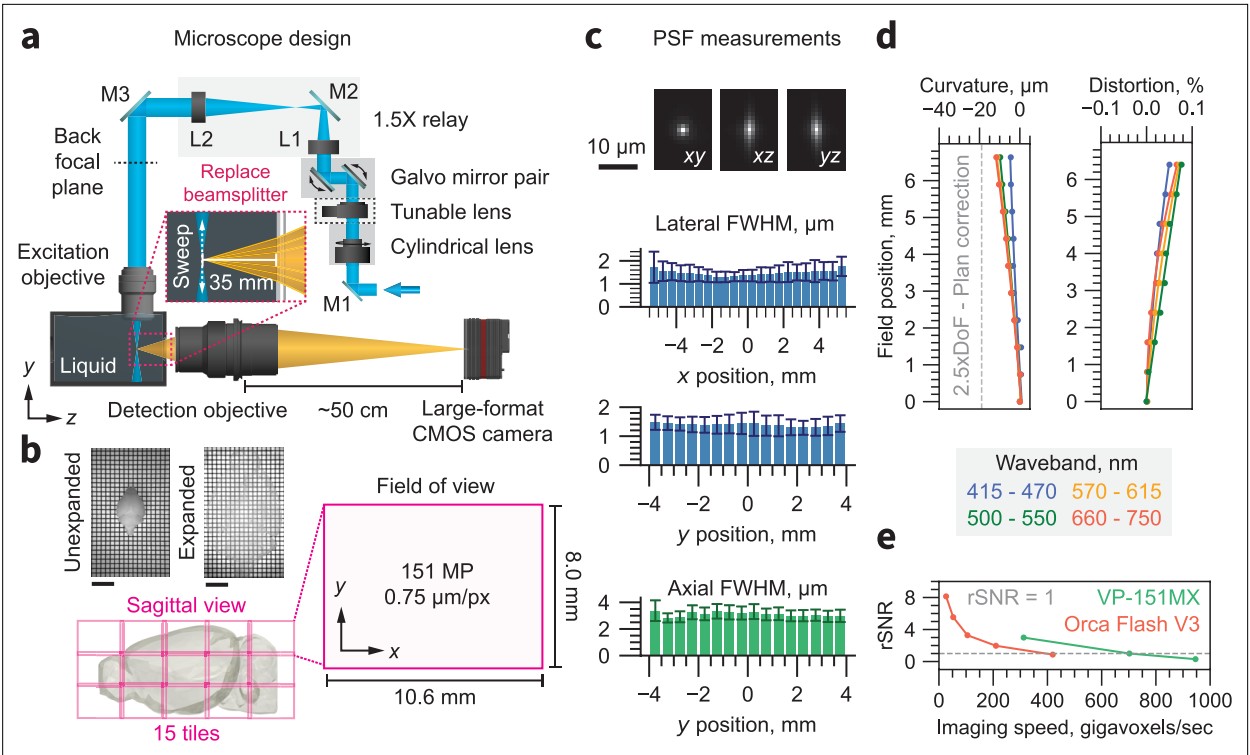

**Figure 2.** Microscope overview. (**a**) Schematic of the ExA-SPIM system. Light enters the system from the laser combiner and is reflected by mirror M1. A cylindrical lens focuses the light in one dimension onto the surface of a tunable lens, which is magnified onto the back focal plane of the excitation through a 1.5×relay consisting of lenses L1 and L2 and mirrors M2 and M3. The excitation objective is oriented vertically and dipped into a liquid immersion chamber. The tunable lens is conjugated to the back focal plane of the excitation objective to enable axial sweeping. A pair of galvo mirrors is used in tandem to translate the position of the light sheet in z (along the optical axis of the detection objective). The detection objective is oriented horizontally. A beam splitter is removed from the lens and replaced with approximately 35 mm of water. A large-format CMOS camera captures images from the detection lens, at a back focusing distance of 50 cm. (**b**) The field of view of the system is 10.6×8.0 mm (13.3 mm diagonal), which is digitized by the camera into a 151-megapixel (MP) image with 0.75 µm/px sampling. Although the optical resolution of the detection lens is ~1.0 µm, the sampling limited resolution based on the Nyquist criterion is ~1.5 µm. The large field of view dramatically reduces the need for tiling. For example, a 3× expanded mouse brain can be captured in only 15 tiles. Representative images of a three expanded mouse brain are shown with a 1 cm scale bar. (**c**) The PSF for 561 nm excitation is shown in the xy, xz, and yz planes. The mean and standard deviation of the lateral and axial full-width half-maximum are shown as a function of x and y position across the full field of view. (**d**) The field curvature and distortion of the system as a function of field position is shown for different wavebands. The field curvature is <2.5× the depth of field (DoF) for all wavebands. This performance is better than 'Plan' specified life sciences objectives (*Zhang and Gross, 2019a*). (**e**) The relative signal-to-noise ratio (rSNR) of the VP-151MXCMOS camera and an Orca Flash V3 sCMOS camera as a function of imaging speed. The VP-151MX camera provides equivalent SNR at nearly twice the imaging speed.

The online version of this article includes the following figure supplement(s) for figure 2:

**Figure supplement 1.** Comparison of traditional cleared tissue SPIM and ExA-SPIM imaging of a mouse brain at equivalent resolutions.

**Figure supplement 2.** Comparison of traditional scientific and electronics metrology technologies.

**Figure supplement 3.** Theoretical noise characteristics of sCMOS and Sony IMX411 sensors.

**Figure supplement 4.** CAD renderings of the microscope detailing (**a**) the complete system, (**b**) the detection assembly, (**c**) the illumination assembly, (**d**) the chamber assembly, and (**e**) the stage assembly.

**Figure supplement 5.** The system uses three 1000 mW lasers at 488, 561, and 639 nm (Genesis MX-STM series, Coherent) with an optional 405 nm laser.

**Figure supplement 6.** The same region in an expanded mouse brain sample was irradiated repeatedly to measure photobleaching at the excitation powers used by the ExA-SPIM.

**Figure supplement 7.** ExA-SPIM imaging of cleared mouse brains.

**Figure supplement 8.** Summary plot comparing the resolution, isotropy, and imaging speed of various existing large volume microscopy methods (2p and SPIM).

**Figure supplement 9.** A block diagram summarizing the acquisition procedure for an ExA-SPIM dataset is shown.

**Figure supplement 10.** Although clearing and expansion protocols render tissues transparent, there still exist small refractive index inhomogeneities, ΔRI, which degrade image quality at larger imaging depths.

Based on these lenses, and the travel limits of a mechanical motorized stage, the ExA-SPIM is capable of imaging a 200×52 × 35 mm$^3$ volume, with ~1.5 µm lateral and ~3 µm axial native optical resolution, and minimal field curvature quantified using the methodology described in *Vladimirov et al., 2024* and distortion (*Figure 2c–d*). We leveraged this large volume to image intact, expanded tissues.

With 3×tissue expansion, the system can image a native tissue volume of 67×17 × 12 mm$^3$ with an effective optical resolution of ~0.5 µm laterally and ~1 µm axially. This amounts to >100 teravoxels, which can be captured without the need for physical sectioning and with minimal tiling. The large field of view of the ExA-SPIM system can also permit imaging cleared tissues, such as mouse brains, in a single tile (*Figure 2—figure supplement 7*). A summary plot comparing the focal volume, isotropy, and imaging speed of our new ExA-SPIM system to other large-scale volumetric imaging systems is shown in (*Figure 2—figure supplement 8*). Individual data points are listed in *Appendix 1—table 2*.

The large-scale and high-speed imaging made possible by the ExA-SPIM system required new software for controlling the microscope, as well as downstream computational pipelines for compressing and handling the resulting datasets (*Figure 2—figure supplement 9*). We developed custom acquisition software capable of interfacing with the VP-151MX camera driver (https://github.com/acquire-project), which streams imaging data directly to next-generation file formats (*Beati et al., 2020*; *Clack et al., 2023*; *Moore et al., 2021*). Using a combination of high-speed networking, fast on-premises storage, and real-time compression, our current imaging pipeline enables a throughput to cloud storage of >100 TB per day, using commodity hardware. This pipeline is actively being developed, with ongoing efforts focused on standardizing every processing step around the OME-Zarr file format (*Moore et al., 2023*).

## Whole-brain expansion

Tissue expansion for microscopy (ExM) enables imaging optically transparent specimens at effective resolutions well below the diffraction limit of light microscopes (*Chang et al., 2017*; *Chen et al., 2015*; *Tillberg and Chen, 2019*; *Figure 2—figure supplement 10*). Moreover, ExM can produce optically clear specimens with low fluorescence background. Multiple ExM variations have emerged, driven by specific biological questions. These include engineered hydrogel chemistry for post-expansion molecular interrogation of proteins or RNA (*Asano et al., 2018*; *Cho and Chang, 2022*; *Tillberg et al., 2016*; *Cui et al., 2022*), formulations that provide gel stiffness (*Chen et al., 2021*) or tunable expansion up to >10 × (*Damstra et al., 2022*; *Klimas et al., 2023*; *Sarkar et al., 2022*; *Truckenbrodt et al., 2018*). Most of these protocols have been developed for specimens that are at most 100 micrometers thick. We developed ExM methods for centimeter-scale tissue samples, including entire mouse brains. A key requirement for ExA-SPIM is optical clearing so that the entire volume can be imaged with diffraction-limited resolution without sectioning. Index of refraction inhomogeneities in heavily myelinated fiber tracts pose particular challenges. Clearing was achieved by stringent dehydration and delipidation prior to gelation and expansion. We systematically evaluated dehydration agents, including methanol, ethanol, and tetrahydrofuran (THF), followed by delipidation with commonly used protocols on 1-mm-thick brain slices. Slices were expanded and examined for clarity under a macroscope. Dehydration using THF was followed by two sequential delipidation steps. First DISCO type clearing *Chi et al., 2018*; *Renier et al., 2014* followed by aqueous delipidation (*Chen and Svoboda, 2020c*, dx.doi.org/10.17504/protocols.io.zndf5a6) rendered the samples extremely transparent (*Figure 3a*; *Ouellette et al., 2023*, dx.doi.org/10.17504/protocols.io.n92ldpwjxl5b/v1). In addition to clearing specimens, delipidation facilitates immunolabeling brain samples prior to gelation and expansion (*Figure 3*). Signal amplification facilitates high contrast imaging of small structures, especially since the increase in volume upon expansion dilutes the concentration of the fluorophore. For gelation, we used VA-044 as the initiator, instead of the more commonly used APS/TEMED (*Tillberg and Chen, 2019*). VA-044 initiates free radicals at a temperature-dependent rate. At low temperatures (4 °C), the gelling reagents are allowed to diffuse to the center of thick samples, followed by higher temperatures (37 °C) to trigger uniform polymerization. Our protocol provides nearly isotropic expansion of the whole sample (including internal brain structures). Although developed for the mouse brain, we have successfully used this protocol for other large specimens, such as a 1 cm×1 cm×1.5 cm piece of macaque motor cortex, as well as a 1 cm×1 cm×0.01 cm section of human visual cortex (*Ouellette et al., 2023*, dx.doi.org/10.17504/protocols.io.n92ldpwjxl5b/v1).

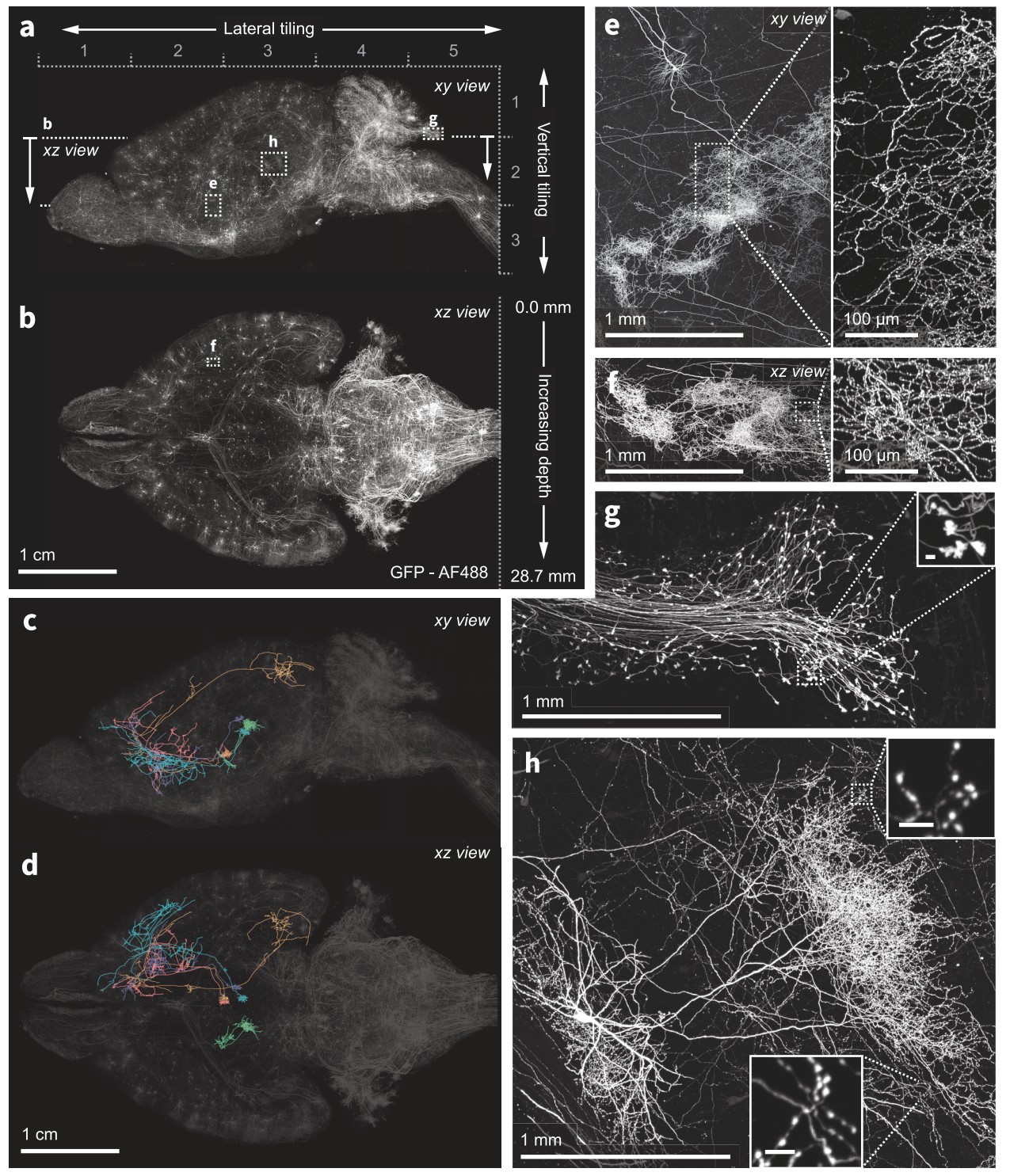

**Figure 3.** Nanoscale imaging of intact mouse brains. (**a–b**) Neurons in the mouse brain were labeled using a combination of PHP.eB pseudotyped enhancer AAV (pAAV-AiE2255m-minBG-iCre(R297T)-BGHpA that drives expression of a mutated iCre(R297T) recombinase and targets the 149 PVT-PT Ntrk1Glut molecular subclass in the whole mouse brain taxonomy *Yao et al., 2023*) together with an AAV expressing GFP in a Cre-dependent manner. Sparse and bright labeling was achieved for neurons in the olfactory bulb, striatum, thalamus, midbrain, and medulla. Intact mouse brains were expanded (3×) and imaged using the ExASPIM microscope (*Ouellette et al., 2023*, dx.doi.org/10.17504/protocols.io.n92ldpwjxl5b/v1). (**c–d**) Representative reconstructions of complete axonal morphologies of five thalamic neurons. (**e–h**) ExA-SPIM imaging resolves dense axonal arbors and varicosities across various brain regions with high resolution and isotropy . *xy* and *xz* views of a dense arbor in the striatum (**e–f**); mossy fiber

*Figure 3 continued on next page*

*Figure 3 continued*

axon terminals and boutons (**g**); an individual hippocampal CA3 neuron and its extensive local axons (**h**). Images are displayed as maximum intensity projections, and inset scale bars correspond to 10 μm post tissue expansion unless otherwise specified.

The online version of this article includes the following figure supplement(s) for figure 3:

**Figure supplement 1.** Representative regions of interest of individual axons in a mouse brain are shown at increasing imaging depths.

## Imaging single neurons across entire mouse brains

We cleared and expanded entire mouse brains and imaged individual neurons, resolving their dense axonal projections (*Figure 3*). Tracking the axons of individual neurons is a challenging problem—axon collaterals can be very thin (<100 nm) and traverse large distances (centimeters), spanning vast areas of the brain. This requires high-resolution, high-contrast imaging of the entire brain without loss of data. Current best-in-class approaches (*Economo et al., 2016*; *Gong et al., 2013*; *Gong et al., 2016*; *Winnubst et al., 2019*) require physical sectioning and extensive tiling, which complicates downstream data processing. In addition, imaging lasts multiple days, which increases the chance for experimental failure and data loss. Finally, the resolution of these imaging methods is highly anisotropic (typically <1:6), which compromises the ability to perform unambiguous axon tracing.

We expressed GFP in a sparse subset of neurons in an enhancer virus mouse to label subcortical projection neurons. Brains were expanded (3×) and imaged in 15 tiles in as little as 24 hr. Because of the lack of physical tissue slicing and minimal tiling, datasets are aligned with an average residual error of <2 pixels between interest point correspondences in neighboring tiles (*Hörl et al., 2019*). Tile alignment of a representative dataset based on initial stage coordinates (*Video 1* and *Video 2*), after stitching optimization (*Video 3*), and after fusion (*Video 4*) are available as videos. Dense axonal projections and varicosities are clearly visible (*Figure 3e–h*), and long-range axons can be tracked across the brain (*Figure 3c–d*). Videos of the data displayed in *Figure 3e and f* are shown in *Video 5 Video 6*.

Representative images of axons as a function of imaging depth are shown in (*Figure 3—figure supplement 1*). The achromatic performance of the ExA-SPIM optics also enables multiple fluorescence channels to be acquired from mouse brains (e.g. GFP and tdTomato; *Figure 4*).

## Imaging cortico-spinal tract neurons in macaque motor cortex

We next established that our methods are applicable to larger brains. With minor adaptations, we applied our clearing and expansion protocol to a 1 cm×1 cm×1.5 cm block of pigtail macaque brain from the hand-wrist and trunk regions of primary motor cortex. Due to its strong myelination, this region of the primate brain is particularly difficult to clear and image (*Van Essen et al., 2019*). The specimen contained cortico-spinal neurons expressing a fluorescent protein (tdTomato) driven by retrograde AAV injected into the intermediate and ventral laminae of the cervical spinal cord. ExA-SPIM image volumes revealed brightly labeled cortico-spinal neurons (*Figure 5a–b* and *Video 7*, *Video 8*, *Video 9*). Individual neurons, their dendritic arbors, and extensive dendritic spines as well as the descending axon and collaterals are clearly discernible (*Figure 5c–f* and *Video 10*). Additionally, even in down-sampled data (~1 μm effective voxel size), we can follow axonal pathways. This raises the possibility that with a small number of slices (~6

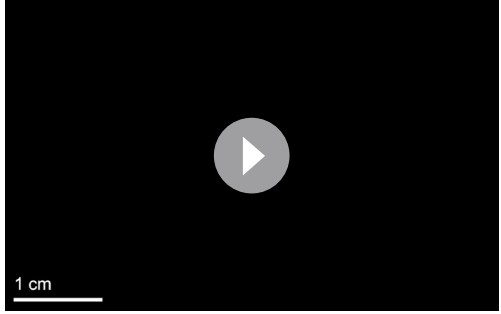

**Video 1.** Tiled imaging of an expanded mouse brain. Adjacent tiles are colored magenta and green.
https://elifesciences.org/articles/91979/figures#video1

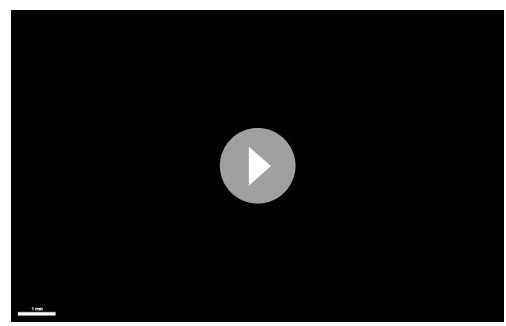

**Video 2.** Zoom-in of a single tile before tile alignment.
https://elifesciences.org/articles/91979/figures#video2

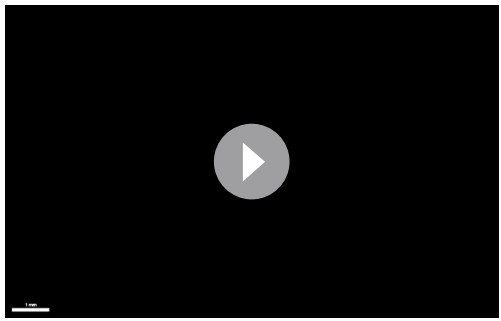

**Video 3.** Zoom-in of a single tile after tile alignment.
https://elifesciences.org/articles/91979/figures#video3

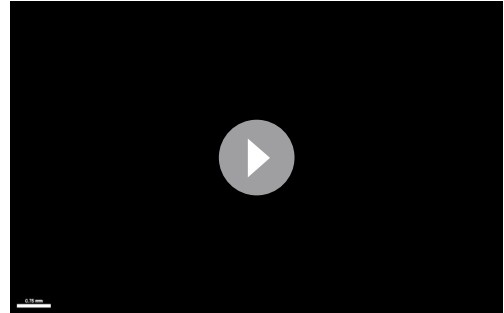

**Video 4.** Zoom-in of a single tile after tile fusion.
https://elifesciences.org/articles/91979/figures#video4

× 1 cm thick slabs) and reduced imaging resolution, a modified inverted (*Wu et al., 2013*; *Kumar et al., 2014*) or open-top (*McGorty et al., 2015*; *Glaser et al., 2017*; *Glaser et al., 2019*; *Barner et al., 2020*; *Glaser et al., 2022*) ExA-SPIM design (*Figure 5—figure supplement 1*) could provide a mesoscale map of white matter axons across the entire macaque brain in several days. Existing efforts to map pathways in the primate brain at this resolution require slicing the brain in 300 µm sections and laboriously assembling the resulting images into a coherent 3D volume (*Xu et al., 2021*).

## Visualizing axons in human neocortex and white matter

Heavy chain neurofilaments comprise the internal scaffolds of long-range projection axons. Visualizing these neurofilaments with immunofluorescence could provide detailed information about axonal trajectories without the need for gene transfer methods, and which cannot be achieved with other methods, such as diffusion magnetic resonance imaging (*Huang et al., 2021*; *Van Essen et al., 2019*) or high-resolution optical coherence tomography (*Li et al., 2019*). We next evaluated ExA-SPIM for imaging immunolabeled heavy chain neurofilaments in the human neocortex (*Figure 6*).

We cleared and expanded (4×) a~1 cm×1 cm×0.01 cm piece of human neocortex (*Figure 6a*) and labeled the tissue with fluorescent SMI-32, which preferentially stains heavy chain neurofilaments. Individual axons and their trajectories are clearly visible through the 350-µm-thick sample (*Figure 6b–d*). The larger axons (>1 µm) are well separated from each other in both the gray and white matter. In the white matter, axons are arranged as multiple intercalated populations coursing in different directions, rather than as homogeneous fascicles. These results demonstrate the feasibility of using ExA-SPIM for visualizing white matter tract axons and lay the foundation for scaling this data acquisition to multiple, thick tissue sections, and ultimately the entire human brain.

## Discussion

Recent breakthroughs in histological methods (*Richardson and Lichtman, 2015*; *Spalteholz, 1914*; *Dodt et al., 2007*; *Tsai et al., 2009*; *Hama et al., 2011*; *Becker et al., 2012*; *Ertürk et al., 2012*; *Chung and Deisseroth, 2013*; *Ke et al., 2013*; *Susaki et al., 2014*; *Tainaka et al., 2014*; *Yang et al., 2014*; *Renier et al., 2014*; *Hou et al., 2015*; *Costantini et al., 2015*; *Chi et al., 2018*), fluorescent labeling strategies (*Kim et al., 2015*; *Susaki et al., 2020*; *Mao et al., 2020*), fast and sensitive cameras, and new microscopes (*Huisken et al., 2004*; *Dodt et al., 2007*; *Wu et al., 2013*; *Kumar et al., 2014*; *Tomer et al., 2014*; *Economo et al., 2016*; *Narasimhan et al., 2017*; *Power and Huisken, 2017*; *Migliori et al., 2018*; *Chakraborty et al., 2019*; *Glaser et al., 2019*; *Voigt et al., 2019*; *Voleti et al., 2019*; *Chen et al., 2020b*; *Chen et al., 2020a*; *Wang et al., 2021*; *Xu et al., 2021*; *Zhang et al., 2021*; *Glaser et al., 2022*; *Qi et al., 2023*; *Vladimirov et al., 2024*) are revolutionizing our ability to study the molecular organization of cells and tissues with fluorescence microscopy.

For many applications, it is necessary to reconstruct tissues with high resolution and contrast over large spatial scales, including cells, organs, or even entire organisms. Such multi-scale imaging is necessary to reconstruct individual neurons in the mouse brain, which can span many millimeters, but with axons that are often less than 100 nm thick. ExA-SPIM addresses this need for imaging

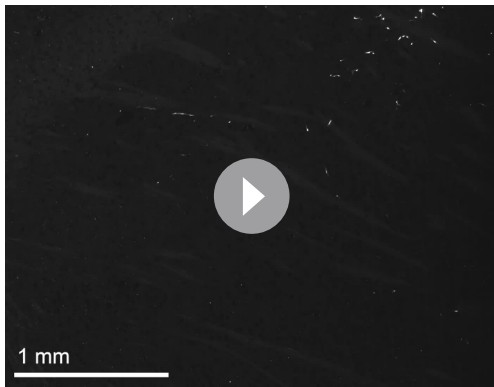

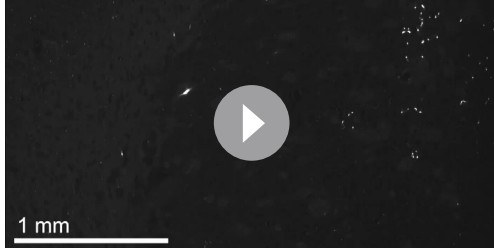

**Video 6.** XZ fly through of dense axonal arbors shown in *Figure 3f*.
https://elifesciences.org/articles/91979/figures#video6

**Video 5.** XY fly through of dense axonal arbors shown in *Figure 3e*.
https://elifesciences.org/articles/91979/figures#video5

of large tissue volumes with high resolution and contrast. The resulting image volumes are sufficient for manual neuron tracing, and preliminary data suggests that automated segmentation of thin axons is improved compared to previous approaches (*Winnubst et al., 2019*).

ExA-SPIM relies on tissue expansion and lower NA optics compared to microscopes that are typically used for high-resolution imaging. This approach has several advantages over imaging of cleared (non-expanded) tissues. First, optical engineering requirements are more forgiving for lenses with lower NA. As shown in (*Figure 1b*), lower NA lenses can have much higher etendues, fields of view, and much longer working distances. In other words, a large volumetric coverage (accessible resolvable voxels) is more feasible with lower NA lenses. In addition, these lenses are more easily corrected for common distortions, including field curvature, pincushion distortion, vignetting, and uniform resolution across the entire field of view (*Zhang and Gross, 2019a*). These lenses can be paired with large format CMOS sensors that offer voxel rates surpassing more commonly used scientific CMOS sensors. These technologies will continue to progress rapidly. For example, the recently announced 247-megapixel Sony IMX811 sensor will be capable of delivering 12-bit images at up to 3 gigavoxels/s.

Second, lower NA imaging is less sensitive to tissue-induced aberrations. Although clearing and expansion techniques render tissues transparent, remnant refractive index inhomogeneities still deteriorate image quality at larger imaging depths (*Weiss et al., 2021*). This is more noticeable at higher NA, because the rays entering the objective at higher angles have longer path lengths through the tissue and are more susceptible to aberrations (*Figure 2—figure supplement 4*). For lower NA systems, the differences in path lengths between the extreme and axial rays are smaller. For example, the effect of spherical aberration (e.g. loss of Strehl ratio, which may be a dominant aberration mode for imaging cleared and expanded tissues) increases with NA to the third power (*Hecht, 2015*).

Third, the refractive index gradients in tissue are expected to decrease with the third power of the expansion factor (proportional to density) (*Jacques, 2013*). As a consequence, tissue-induced aberrations reduce with the expansion factor to the third power. In contrast, the imaging path length through the tissue only increases with the expansion factor to the first power. These factors together may explain the advantage of expansion-assisted imaging (*Figure 2—figure supplement 10*). Further investigation into the optical properties of hydrogels and their constituents is necessary to fully support these hypotheses. ExA-SPIM still exhibits a decrease in image quality at larger imaging depths (*Figure 2—figure supplement 1*). This can be reduced with dual-sided excitation (*Dodt et al., 2007*) as well as multi-view detection (*Tomer et al., 2012*).

One limitation of the current implementation of the ExA-SPIM is related to inefficient collection of signal photons. Given that signal collection scales approximately with $NA^2$, signal collection is 10×lower compared to a NA = 1.0 system ($(1/0.305)^2$). However, we find that the ExA-SPIM system can detect and resolve dim, nanoscale biological features with high signal-to-noise ratio (e.g. thin axons). Because of the large field of view, high SNR imaging requires 1000+mW lasers. However, light intensities are similar to traditional SPIM systems because the power is distributed over a ~1 cm wide light sheet. Under these imaging conditions, photobleaching is not detrimental (i.e. ~50% reduction in intensity after ~800 repeated exposures), suggesting that even higher laser intensities could be used for higher SNR imaging (*Figure 2—figure supplement 6*). To improve signal collection and

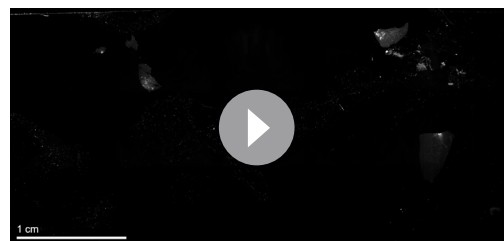

**Video 7.** XY fly-through of macaque motor cortex.
https://elifesciences.org/articles/91979/figures#video7

enable even higher sensitivity ExA-SPIM imaging, a custom lens with NA = 0.5, FOV = 6.8, and a 35 mm working distance in liquid has been designed and is being fabricated. This new lens will enable ~50–150 nm effective lateral resolution in 3–12×expanded tissues, with 2.7×improved light collection efficiency.

Fundamentally, contrast, effective resolution, and imaging speed in expansion microscopy are limited by the density of fluorescent molecules labeling the structure of interest and the photon budget. ExA-SPIM allows distributing the technical burden between optics and tissue expansion (*Figure 1—figure supplement 1*). In practice, a target resolution should first be specified, corresponding to a viable combination of NA and tissue expansion, taking into consideration light collection efficiency (NA dependent) and required working distance (expansion factor dependent).

Although we have focused on the combination of new microscopy with tissue expansion, the ExA-SPIM microscope design will be useful for many additional imaging applications. For example, the field of view of the ExA-SPIM is sufficient to image a cleared mouse brain without the need for tiling. Cleared mouse brain datasets with ~1.5 µm lateral resolution could be acquired at a speed of ~36 min/channel (*Figure 2—figure supplement 7*). In addition, the entire system could be converted to an inverted (*Wu et al., 2013*; *Kumar et al., 2014*) or open-top (*McGorty et al., 2015*; *Glaser et al., 2017*; *Glaser et al., 2019*; *Barner et al., 2020*; *Glaser et al., 2022*) architecture, which would enable large-scale imaging of tissue slabs with large aspect ratios that are up to 1 cm thick (*Figure 2—figure supplement 1*). Similarly, an ExA-SPIM system with a lower resolution metrology lens could easily be built (see *Appendix 1—table 3*). These more mesoscale systems would provide extremely high imaging throughput (cm³ per day) and could pave the way for new large-scale neuroanatomy investigations of human and non-human primate tissues.

Finally, new types of applications may also require multi-scale microscopy. For example, expansion microscopy could provide a super-resolution view of interactions between immune cells and solid tumors. This application may require 10×expansion, for better than 100 nm effective resolution, to distinguish immunofluorescence in immune cells and tumor cells, throughout millimeter-scale tissue samples (10 mm after expansion). Given that 10×expanded samples are mechanically fragile and difficult to handle and section, the large field of view and reduced need for physical sectioning of ExA-SPIM would be highly advantageous for these demanding imaging experiments.

In summary, our new ExA-SPIM approach represents a new substrate for innovation within the realm of large-scale fluorescence imaging. By leveraging technologies from the electronics metrology industry, in combination with whole-mount tissue expansion, the system provides: (1) nanoscale lateral resolution, (2) over large centimeter-scale volumes, (3) with minimal tiling and no sectioning, (4) high isotropy, and (5) fast imaging speed.

## Materials and methods
### Microscope design
A parts list and CAD model are available at https://github.com/AllenNeuralDynamics/exa-spim-hardware (*Allen Institute for Neural Dynamics, 2023b*). A ZEMAX model is available at https://github.com/AllenNeuralDynamics/exa-spim-optics (*Allen Institute for Neural Dynamics, 2023a*). The system has two main optical paths (*Figure 2—figure supplement 4*). The first excitation path shapes and delivers the light sheet to the specimen. Laser light is provided by a custom-made laser combiner containing a 200 mW

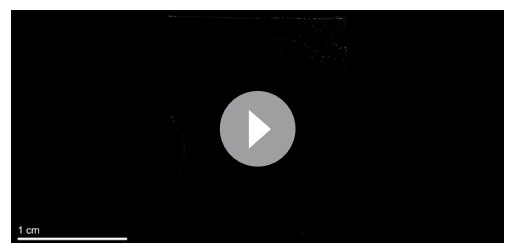

**Video 8.** YZ fly-through of macaque motor cortex.
https://elifesciences.org/articles/91979/figures#video8

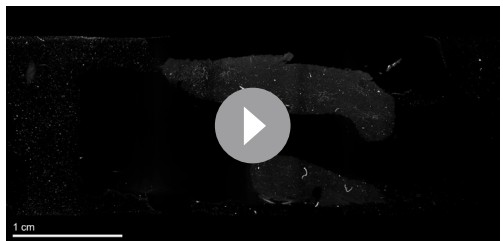

**Video 9.** XZ fly-through of macaque motor cortex.
https://elifesciences.org/articles/91979/figures#video9

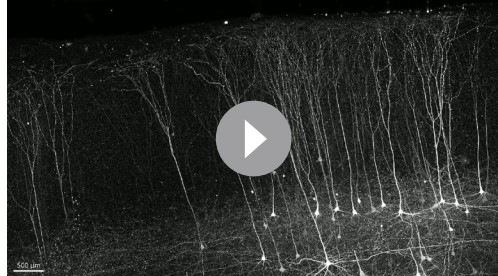

**Video 10.** Tracking cortico-spinal tract neurons in macaque motor cortex.
https://elifesciences.org/articles/91979/figures#video10

405 nm laser (LBX-405–180-CSB-OE, Oxxius), 1000 mW 488 nm laser (Genesis MX488-1000 STM OPS Laser-Diode System, Coherent Inc), 1000 mW 561 nm laser (Genesis MX561-1000 STM OPS Laser-Diode System, Coherent Inc), and 1000 mW 639 nm laser (Genesis MX639-1000 STM OPS Laser-Diode System, Coherent Inc; *Figure 2—figure supplement 5*). An acousto-optic tunable filter (AOTFnC-400.650-TN, AA Opto Electronic) controlled by a RF driver (MPDS8C-D65-22-74.158-RS, AA Opto Electronic) is used to modulate each laser and control the output power of the combiner. The output beam from the combiner is ~1.2 mm in diameter, which is expanded to 13.3 mm using two relay lenses with $f$=3.6 mm (LMPLANFLN 50×, Olympus) and $f$=40 mm (HLB M PLAN APO 5×, Shibuya Optical). The light path is then reflected vertically by a third kinematic mirror and focused along one axis by an achromatic cylindrical lens (ACY254-050-A-ML, Thorlabs). The cylindrical lens is mounted in a motorized rotation mount (C60-3060-CMR-MO, Applied Scientific Instrumentation). This enables precise electronic control over the rotation of the light sheet within the specimen. Control over this parameter is important, as the field of view (~10.6 mm) and depth of field of the detection lens (<10 μm) require the sheet to be rotated with <1 deg precision. The light from the cylindrical lens is focused onto an electrically tunable lens (EL-16–40-TCVIS-20D-C, Optotune AG). The actuating surface of the electrically tunable lens is conjugated to the back focal plane of the excitation objective (1–290419, Navitar) through two large 20 mm galvanometric scanning mirrors (QS20X-AG, Thorlabs), three additional kinematic mirrors, and a final relay consisting of a first lens with $f$=200 mm (AC508-200-A-ML) and second lens with $f$=300 mm (AC508-300-A-ML). The two galvanometric mirrors are not conjugated to the back focal plane of the excitation objective and are used in tandem to tilt and translate the light sheet to be co-planar with the detection lenses' focal plane. Similar to the cylindrical lens rotation, the light sheet must also be tilted to <1 deg precision across the ~8.0 mm vertical height of the imaging field of view.

The excitation objective has a focal length of $f$=110 mm, NA = 0.133, and back aperture diameter of ~29.26 mm, with which the preceding excitation optics results in a Gaussian excitation light sheet with NA ~0.1. The intensity profile across the width of the light sheet is also Gaussian, with a full-width half-maximum (FWHM) of ~12.5 mm. This corresponds to a reduced (60%) light intensity at the edges of the field of view. The lens is mounted in a custom dipping cap which extends the effective working distance from 39 mm in air to ~52 mm in water. The 1-mm-thick window on the dipping cap was sealed with UV curing optical adhesive (NOA86, Norland Products). The lens is oriented vertically, such that the light sheet is delivered downward into the immersion chamber which is filled with an immersion medium.

Fluorescence is collected with a high etendue electronics metrology lens (VEO_JM DIAMOND 5.0×/F1.3, Vieworks Co., LTD; jointly developed and fabricated by Schneider-Kreuznach). Although the spherical aberration introduced in the excitation path is negligible, the same dipping cap design would result in severe spherical aberration on the detection path, where the aperture was larger (NA = 0.305). Imaging at these higher apertures necessitated imaging lenses designed for direct immersion into a liquid mounting medium. The VEO_JM DIAMOND detection lens is designed for co-axial illumination, involving a 35-mm-thick BK7 glass (n=1.52) beam splitter mounted on the front object side of the lens. The beam splitter couples white light into the lens for bright illumination of electronic components during inspection. This was leveraged for aberration-free imaging into a liquid mounting medium.

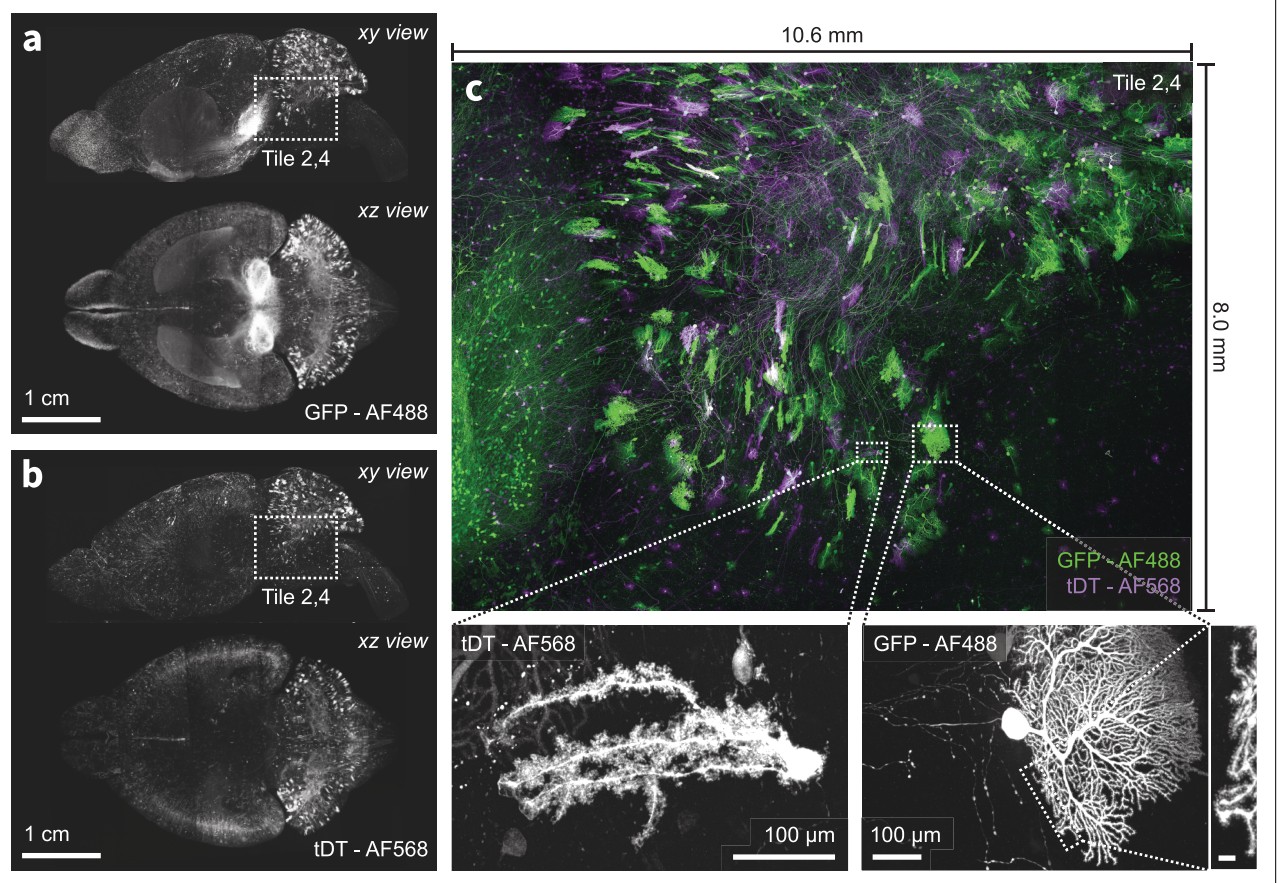

**Figure 4.** Multi-color imaging of centimeter-scale tissues with nanoscale resolution. (**a–b**) An intact mouse brain was expanded (3×) and sparsely labeled neurons expressing GFP and tdTomato were imaged using the ExA-SPIM microscope (**Ouellette et al., 2023**, dx.doi.org/10.17504/protocols. io.n92ldpwjxl5b/v1). An overlay of both channels for a single tile is shown in (**c**). The achromatic performance of the optics enabled diffraction-limited imaging in both color channels, as illustrated by the ability to resolve individual spines along the dendrites of Purkinje cells in the cerebellum. Images are displayed as maximum intensity projections, and inset scale bars correspond to 10 μm post tissue expansion unless otherwise specified.

We removed the beam splitter and replaced it with an optically equivalent thickness of the aqueous mounting medium (30–35 mm for refractive indices between 1.33–1.56) and glass (4 mm total, including the chamber window and fluorescence filter). This scheme avoids the spherical aberrations that would otherwise plague imaging with an air-immersion lens into a non-air-based medium. The lens is positioned horizontally on the optical table outside of the immersion chamber. The lens images the specimen through a $\lambda/10$ 50 mm diameter VIS-EXT coated fused silica window that is 3 mm thick (#13–344, Edmund Optics). The window was glued to the immersion chamber using UV curing optical adhesive (NOA86, Norland Products). A custom 44 mm diameter 1-mm-thick multi-bandpass fluorescence filter (ZET405/488/561/640mv2, Chroma) is inserted on the outer side of the immersion chamber and held in place with a retaining ring (SM45RR, Thorlabs). Placing the filter on the object side of the lens results in a cone angle of incidence of 17.8 deg. This results in a spectral shift and broadening of transmitted light, which was modeled to account for adequate suppression of scattered light at the excitation laser wavelengths.

The lens is attached to the camera (VP-151MX-M6H00, Vieworks Co., LTD) using a custom mechanical assembly that enables tipping and tilting of the lens and camera relative to the immersion chamber window. To reduce fixed pattern noise at low light levels, we requested that the manufacturer permanently disable the photo response non-uniformity (PRNU) correction mode. The camera uses the Sony IMX411 sensor, which features a 14192×10,640 array of 3.76 μm pixels. The ×5.0 magnification of the detection lens implies sampling at 0.75 μm in the specimen plane. Because the detection lens is not infinity-corrected, it is not possible to change tube lenses to change the pixel sampling. Therefore, although the optical resolution of the detection lens is ~1.0 μm, the sampling limited resolution based

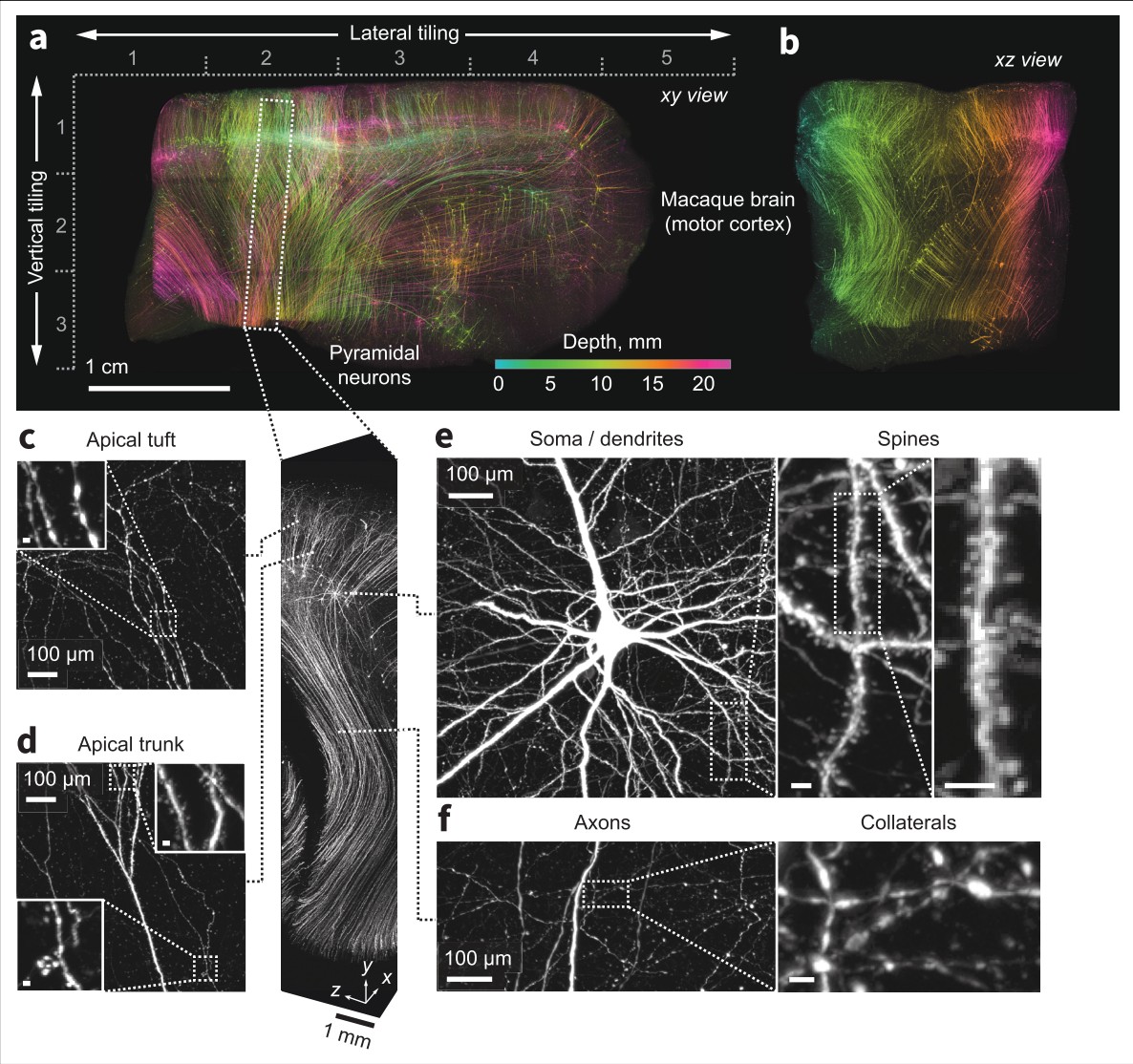

**Figure 5.** Expansion and imaging of a large volume of macaque brain. A 1 cm×1 cm×1.5 cm block of macaque primary motor cortex was expanded (3×) and imaged on the ExA-SPIM (*Videos 7–9*). Corticospinal neurons were transduced by injecting tdTomato-expressing retro-AAV into the spinal cord. (**a–b**) Maximum intensity projections of the imaged volume pseudo-colored by depth. The axes descriptors in (**a**) indicate the 5×3 tiling used to image this volume. (**c–f**) Fine axonal and dendritic structures including descending axons, collaterals, and dendritic spines are clearly discernible in the images throughout the entire volume. See *Video 10*. Images are displayed as maximum intensity projections with the following thicknesses: (**a**) 23 mm, (**b**) 45 mm, (**c–f**) 1 mm. Inset scale bars correspond to 10 µm post tissue expansion unless otherwise specified.

The online version of this article includes the following figure supplement(s) for figure 5:

**Figure supplement 1.** Schematics for configuring the ExA-SPIM system in an (**a**) inverted or (**b**) open-top architecture.

on the Nyquist criterion is 1.5 µm. However, it is worth noting that Vieworks Co. produces cameras with pixel-shifting technology, which would enable finer sampling if needed. The cooling fan on the camera was replaced with a quieter fan operating at 2400 rpm (Noctua NF-A6x25 FLX, Premium Quiet Fan, 3-Pin, 60 mm, Noctua). Using an alignment laser, the entire assembly was aligned to the window. The entire assembly is also mounted on a rail system (XT95, Thorlabs) that enables precise axial alignment of the assembly relative to the chamber window.

The sensor can be operated with 12, 14, or 16-bit analog to digital (A/D) conversion, each of which provides a trade-off between noise and data throughput. When operated with 14-bit A/D, the line time of the sensor is 20.15 µs. With 14192 pixels per row on the sensor, this corresponds to an imaging speed of $703 \times 10^6$ voxels/s. The line times and imaging speeds at 12 and 16-bit are 15.00 and 45.44 µs respectively, corresponding to 946 and $312 \times 10^6$ voxels/s. This contrasts with a state-of-the-art sCMOS

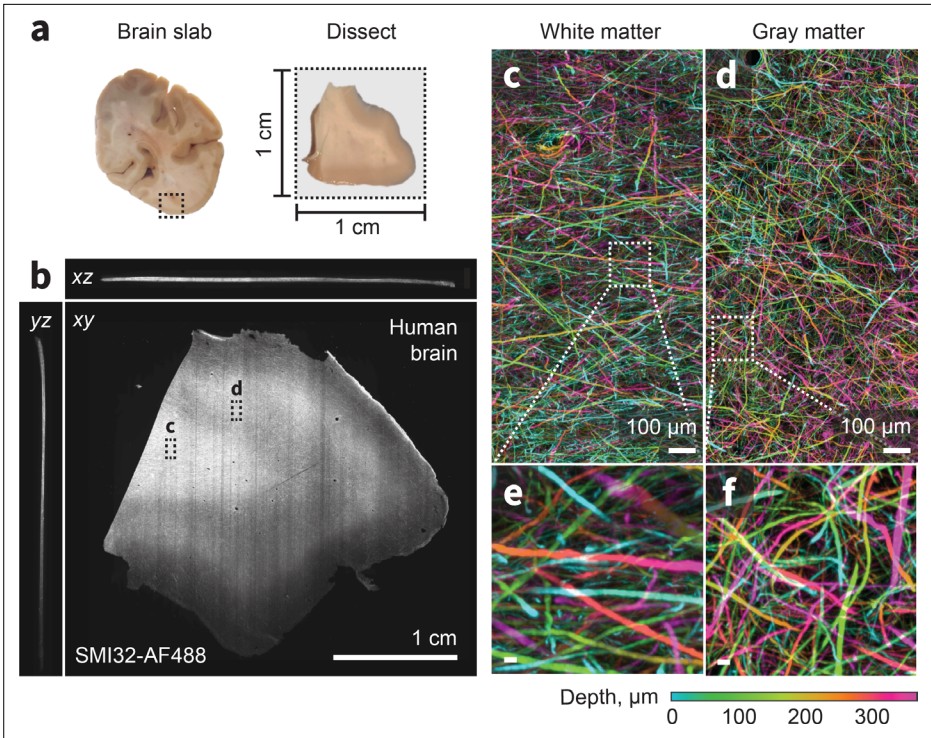

**Figure 6.** ExA-SPIM imaging of human tissue. (**a–b**) A region from the medial temporal-occipital cortex was manually dissected into a ~1 cm×1 cm block, which was subsequently sectioned into ~100 µm sections for tissue expansion (4×), labeling, and ExA-SPIM imaging. (**c–d**) Maximum intensity projection of a region of interest from white and gray matter, pseudo-colored by depth. (**e–f**) Individual axons and their trajectories are clearly resolved with high contrast. Images are displayed as maximum intensity projections across 350 µm (**b–f**). Inset scale bars correspond to 10 µm post tissue expansion unless otherwise specified.

sensor with 2048 pixels per row, where even at the fastest 4.89 µs line time, the imaging speed is only $418×10^6$ voxels/s. In other words, the Sony IMX411 sensor provides more pixel parallelization within each row. This enabled twice the voxel rate with four times the pixel dwell time. See Appendix 1 for further discussion on comparing the speed and sensitivity of cameras.

The specimens, in this case expanded gels, were mounted in a customized holder, which was attached to a motorized XY stage (MS-8000, Applied Scientific Instrumentation) and Z stage (LS-100, Applied Scientific Instrumentation), for scanning and tiling-based image acquisition.

## Microscope control

The microscope was controlled using a high-end desktop workstation (SX8000, Colfax Intl). The workstation was equipped with a motherboard with five PCIe 4.0x16 and one PCIe 4.0x8 slots (X12DAI-N6, Supermicro), most of which are required for the various electronics needed to control the ExA-SPIM. One slot was used for the frame grabber (Coaxlink Octo, Euresys), which streams imaging data from the camera onto a fast local 12.8 TB NVME drive (7450 Max 12800 GB 3 DWPD Gen4 15 mm U.3, Micron). A second slot was used for the data acquisition (DAQ) card (PCIe-6738, National Instruments) used for generating the various digital and analog voltage signals. A third slot was used for the high-speed network interface card (ConnectX-5 EN MCX515A-CCAT QSFP28 Single Port 100GbE, Mellanox) to transfer data off the local NVME drive, over the network, and onto a networked server. A fourth slot was occupied by the workstation GPU (A4000, NVIDIA). The workstation was also connected to a controller (TG-16, Applied Scientific Instrumentation) via a USB connection. The TG-16 controller was equipped with cards for controlling the motorized X, Y, and Z stages, the electrically tunable lens, and the cylindrical lens rotation mount. Finally, each laser within the combiner was connected to the computer via a USB connection, as well as the RF driver of the AOTF. The DAQ acted as the master controller of the entire system. Analog output voltages were used to drive or trigger the electrically

tunable lens, two scanning galvanometric mirrors, camera, scanning stage, and the RF outputs to the AOTF for each laser.

## Acquisition software

The microscope was controlled using custom software written in Python https://github.com/Allen-NeuralDynamics/exaspim-control (*Allen Institute for Neural Dynamics, 2025b*) and was operated on the Windows 10 operating system. The software could control and configure all of the electronically controlled hardware devices and uses Napari https://napari.org as the graphical user interface (GUI) for image streaming and visualization. An imaging experiment consisted of a series of nested loops, which included looping over the total number of frames within a given tile, looping over the total number of channels within a tile, and finally tiling in two dimensions to cover the entire tissue volume. A multiprocessing, double buffering scheme was used to capture a tile while the previous tile was being transferred over the network to longer term storage. All hardware was controlled using a custom library developed for generalized microscope control https://github.com/AllenNeuralDy-namics/voxel (*Allen Institute for Neural Dynamics, 2025a*).

To achieve robust and reliable acquisition at the speed required by the ExA-SPIM, we used two data streaming strategies. The first used the open-source eGrabber frame grabber Python API and open-source Acquire package (https://github.com/acquire-project) (*Clack et al., 2023*), specifically the Python API Acquire-Zarr (*Liddell et al., 2025*). Acquire is a new state-of-the-art microscope acquisition project which enables the ExA-SPIM system to stream data at the required rate, directly to OME-Zarr file format (V2 or V3), with either ZStandard or LZ4 compression, a variable chunk size, a variable shard size (for V3), and an optional multi-resolution pyramid, all of which help streamline downstream data storage and handling. The second strategy used the ImarisWriter library (*Beati et al., 2020*). This option enabled high-speed data streaming with online lossless compression (LZ4 with bit shuffling) and real-time writing of a multi-resolution pyramid.

On average, the lossless compression ratio using either ImarisWriter or Acquire was ~1.5–4× depending on the bit mode of the camera (e.g. 12, 14, or 16 bits). This reduced the overall effective data rate of the system and eased network transfers of the data to centralized storage. A schematic of the acquisition pipeline is shown in *Figure 2—figure supplement 9*.

## Point spread function quantification

To quantify the point spread function of the ExA-SPIM microscope, we imaged fluorescent 0.2 μm TetraSpeck microspheres (Invitrogen, cat no. T7280, lot no. 2427083) using 561 nm excitation. A cube of expanding hydrogel containing a 10% (v/v) microbead solution was prepared, and 40 μL of microspheres solution was added to 360 μL of activated Stock X monomer solution, as described in *Asano et al., 2018*. The solution was briefly vortexed and carefully pipetted into an array of 2 mm$^3$ wells in a silicone mold. The mold was placed in a sealed petri dish containing damp Kimwipes to maintain humidity and incubated at 37 °C for 2 hr. After incubation, the polymerized bead phantoms were placed into 0.05×SSC buffer for at least 24 hr to expand and equilibrate prior to imaging. After volumetric imaging, the resulting imaging stack was analyzed using custom-written Python code https://github.com/AllenNeuralDynamics/exa-spim-characterization (copy archived at *Allen Institute for Neural Dynamics, 2023c*). The results shown in *Figure 2* are averaged from >10,000 individual beads.

## Field curvature quantification

We used the field curvature quantification methods developed for the Benchtop mesoSPIM (*Vladimirov et al., 2024*). Briefly, the field curvature was measured using a high-precision Ronchi ruling with 120 lines per mm (62–201, Edmund Optics). The ruling was mounted into the system and aligned to be normal (i.e. flat) with respect to the imaging path. The ruling was trans-illuminated using light-emitting diodes (LEDs) at 450 nm (M450LP2, Thorlabs), 530 nm (M530L4, Thorlabs), 595 nm (M595L4, Thorlabs), and 660 nm (M660L4, Thorlabs). To provide uniform illumination, the LEDs were collimated to a diameter of ~20 mm and passed through a diffuser (ED1-C20, Thorlabs). Image stacks were captured by scanning the Ronchi ruling in 1 μm steps through the imaging path focal plane (2 mm total scan range) with illumination at either 450 nm, 530 nm, 595 nm, or 660 nm. The resulting image stack was split into a 16×16 grid of regions of interest (ROI), each 887×665 × 2000 pixels. Within each ROI, the contrast of each frame in the stack was calculated using the 5th and 95th percentiles of intensity within

the frame, where the contrast, C = (I$_{max}$ – I$_{min}$)/(I$_{max}$ +I$_{min}$). The resulting contrast versus depth curve was fit to a normal distribution to extract the depth (i.e. index) corresponding to maximum contrast. The indices were radially averaged across the 16×16 grid of ROIs, yielding an estimate of the imaging lens field curvature within each of the four tested wavebands. Custom Python scripts were used to run the analysis https://github.com/AllenNeuralDynamics/exa-spim-characterization.

### Lens distortion quantification

The lens distortion was measured using a target (62–950, Edmund Optics) with 125 µm diameter dots spaced every 250 µm in a grid pattern. The target was mounted and trans-illuminated using LEDs in the same manner as the field curvature quantification. A single image was acquired with the target at the focal plane of the imaging lens. The resulting image was quantified by first segmenting and calculating the centroid of each dot. The calculated position of each dot was then compared to the theoretical dot position. The percent distortion for each dot was defined as the difference between the experimental and theoretical positions, normalized by the theoretical position. The resulting distortion values were radially averaged, yielding the lens distortion as a function of position from the center of the lens field of view. Custom Python scripts were used to run the analysis https://github.com/AllenNeuralDynamics/exa-spim-characterization.

### Camera sensitivity comparison

A bead phantom (see point spread function measurement) was used to compare the sensitivity of the large-format CMOS camera and a scientific CMOS camera (Orca Flash V3, Hamamatsu). A single bead was first imaged with the scientific CMOS camera, varying the effective pixel rate or imaging speed. The scientific CMOS camera was then removed from the microscope and replaced with the large-format CMOS camera. The same bead was located and imaged again at various effective pixel rates. The bead was located in all image stacks, and the signal-to-noise ratio of the bead was quantified as the signal of the bead, minus the average background signal, divided by the background noise. A custom written Python code was used to run the analysis https://github.com/AllenNeuralDynamics/exa-spim-characterization . It is worth noting that the Orca Flash V3 is an older scientific CMOS camera. Newer cameras, such as the Orca BT Fusion with improved noise characteristics (1.0 e- versus 1.6 e- read noise and 95% vs 85% peak quantum efficiency), are now available. However, these differences would not significantly change the results shown in *Figure 2e*. Appendix 1 contains a deeper discussion on comparing the speed and sensitivity of cameras.

### Local data storage and handling

Data was streamed using the Acquire or ImarisWriter onto a local NVME drive in the acquisition workstation. Upon the completion of a tile, the resulting files were transferred over the high-speed network using xcopy onto a 670 TB enterprise storage server from VAST data. This occurred in parallel with the acquisition of the next imaging tile. Upon completion of the transfer, the tile was deleted off the acquisition workstation's NVME drive.

### Image compression and cloud transfer

The raw data for each color channel consisted of a set of overlapping 3D image tiles. For the ImarisWriter acquisition workflow, the images were stored in Imaris IMS file format, an HDF5-based format which enables fast parallel writing to local storage. However, this format is not well suited to being archived in cloud (object) storage since accessing an arbitrary data chunk requires seeking a file. Thus, the data was converted to OME-Zarr format, which supports parallel read-write over the network and has a rich metadata structure (*Moore et al., 2021*; *Moore et al., 2023*). OME-Zarr is also integrated with visualization and annotation tools used for downstream visualization and analysis, including Horta Cloud (https://github.com/JaneliaSciComp/hortacloud; *Clements et al., 2025*) and Neuroglancer (https://github.com/google/neuroglancer; *Maitin-Shepard, 2025*; *Maitin-Shepard et al., 2021*). With up to four channels, the storage footprint for a single dataset can reach hundreds of terabytes. Therefore, the storage ratios and compression/decompression speeds of various lossless codecs were compared. Blosc ZStandard yielded the highest storage ratios, with compression speeds comparable to LZ4 at lower 'clevel' settings (e.g. 1–3). Dask was used to parallelize the compression and OME-Zarr writing over a high-performance computing (HPC) cluster consisting of 16 nodes, each with 32

Intel CPUs with Advanced Vector Instructions 2 (AVX2) and 256 GB RAM. Image chunks were read in parallel from high-bandwidth network storage, compressed in memory, and written directly to AWS S3 and Google Cloud Storage buckets. Combined throughput (chunk read, compress, write) reached over 2 GB/s, with execution time dominated by read-write I/O operations. The image compression and cloud transfer code is available at: https://github.com/AllenNeuralDynamics/aind-data-transfer (*Arshadi, 2024*). For the Acquire acquisition workflow, datasets were streamed directly to the OME-Zarr format.

## Image stitching

ExA-SPIM image stitching used a combination of on-premises and cloud-based resources. Datasets were first converted from OME-Zarr to N5 and stitched using BigStitcher on the Google Cloud Platform (GCP) (*Hörl et al., 2019*). Tile placement transformations were based on interest points. The interest point-based tile registration consisted of three steps: identification of interest points, finding corresponding interest points between tiles, and optimizing for tile transformation parameters with respect to the distance of corresponding interest points. We first performed a translation-only stitching routine, followed by a full affine transformation optimization. The affine transformation was regularized by a rigid transformation, and corner tiles were kept fixed to prevent the global scaling of the sample and divergent solutions at the corner tiles. As a rule of thumb, several thousand interest points per overlapping region were required for a reliable identification of correspondences and successful optimization. Once the alignment transformations were calculated, the tiles were fused using the on-premises HPC into a single contiguous volume in the N5 format. A representative dataset (~100 TB in raw uncompressed size) can be fused in ~24 hr using 16 nodes (32 cores each, 480 total), with 16 GB of RAM per core (7.68 TB total) and an output chunk size of 256, 256, 256 pixels. 2 cores per node are reserved as overhead per spark worker (i.e. 32 total for 16 nodes). Fused datasets along with the raw tiled datasets and the tile placement transformations are deposited in an Amazon Web Services S3 aind-open-data data bucket. The N5 datasets were converted to OME-Zarr for visualization with Neuroglancer, or KTX (https://registry.khronos.org/KTX) for visualization with HortaCloud. The current pipeline involves several file format conversion steps and transfers between on-premises and cloud-based platforms. Future pipelines will standardize around OME-Zarr and be completely operable in the cloud. Only the initial data conversion and compression step would be computed on-premises.

## Image visualization and annotation

To generate whole-brain single neuron reconstructions, we use HortaCloud, an open-source streaming 3D annotation platform enabling fast visualization and collaborative proofreading of terabyte-scale image volumes. Human annotators proofread stitched image volumes using HortaCloud in a web browser on a personal workstation. Starting from the soma, the axonal and dendritic arbors were traced through all terminals by laying down connected points along the neurite, producing a piecewise-linear approximation of neuronal trees (*Economo et al., 2016*).

## Whole mouse brain tissue processing

Briefly, the samples were cleared, immunolabeled, and expanded as described below. Detailed protocols for preparing cleared and expanded brains are available at [(*Ouellette et al., 2023*), dx.doi.org/10.17504/protocols.io.n92ldpwjxl5b/v1].

### Viral labeling in mice

Adult transgenic Cre driver mice between ages p21 to p35 received systemic injections, via the retro-orbital sinus, of a 100 µL mixture of Cre-dependent Tet transactivator (AAV-PHP-eB_Syn-FlexTRE-2tTA, typical dose $6.0×10^8$ gc/mL) and a reporter virus (AAV-PHP-eB_7x-TRE-tdTomato, typical dose $1.8×10^{11}$ gc/mL; Addgene plasmid id: #191210 and, #191207). Viral vector-mediated recombination was achieved by retro-orbital injection of AiP1999 - pAAV-AiE2255m-minBG-iCre(R297T)-BGHpA (Addgene id: 223843) along with the two aforementioned vectors. The viral titers of the tTA virus used were empirically adjusted based on the Cre driver line to yield sparsely labeled brains. Viruses were obtained from either the Allen Institute for Brain Sciences viral vector core, the University of

North Carolina, BICCN-Neurotools core and were prepared in an AAV buffer consisting of 1×PBS, 5% sorbitol, and 350 mM NaCl.

## Collection

All experimental procedures related to the use of mice were approved by the Institutional Animal Care and Use Committee (IACUC) protocol (#2416) of the Allen Institute for Brain Science, in accordance with National Institutes of Health (NIH) guidelines. Four weeks after viral transfection, mice (~p70) were anesthetized with an overdose of isoflurane and then transcardially perfused with 10 mL 0.9% saline at a flow rate of 9 mL/min followed by 50 mL 4% paraformaldehyde in PBS at a flow rate of 9 mL/min. Brains were extracted and post-fixed in 4% paraformaldehyde at room temperature for 3–6 hr and then left at 4 °C overnight (12–14 hr). The following day, brains were washed in 1×PBS to remove all traces of excess fixative.

## Delipidation

Whole brain delipidation was performed in two stages. First, brains were dehydrated through a gradient of tetrahydrofuran (THF) in deionized water at 4 °C and then delipidated in anhydrous dichloromethane (DCM) at 4 °C. The brains were rehydrated into water through a gradient of THF and then placed in 1×PBS. Second, whole brains were transferred from 1×PBS to a biphasic buffer (SBiP) for 5 days at room temperature and then rinsed in a detergent buffer (B1n) for 2 days.

## Immunolabeling

Delipidated brains were equilibrated in a detergent buffer (PTxw) and then incubated in PTxw containing the primary antibody (10 μg/brain) at room temperature for 11 days. After thorough washing in PTxw, a solution of the secondary antibody (20 μg/brain) was added for 11 days at room temperature. Brains were then rinsed thoroughly with PTxw and transferred to 1×PBS.

## Gelation and expansion

Immunolabeled brains were equilibrated in MES buffered saline (MBS) followed by incubation in acryloyl-X SE (AcX) at 4 °C on wet ice for 4 days. The AcX solution was then rinsed off with 1×PBS and the brain was then transferred to a solution of StockX activated with VA-044 at 4 °C on wet ice for 4 days. After StockX incubation, whole brains were placed in a polymerization chamber and filled with activated StockX solution. The chamber was sealed with a coverslip, placed in an inert atmosphere of $N_2$, and baked at 37 °C for 4+ hr until hydrogel formation. The hydrogel was then digested with proteinase K for 10+ days, until the tissue cleared. Upon completion of digestion, the brain was rinsed with 1×PBS and expanded in 0.05×saline sodium citrate (SSC) until 3× expansion was achieved.

## Macaque brain tissue processing

The process for preparing the macaque brain samples is described below.

### Spinal injection procedure and motor cortex collection

To retrogradely label corticospinal neurons in the hand-wrist and trunk regions of primary motor cortex in a pigtail macaque, we injected a retro AAV vector (rAAV2-CAG-tdTomato; Addgene plasmid #59462 packaged in-house, titer of $1.88×10^{13}$) into the left lateral funiculus and ventrolateral part of the gray matter in the C6/C7 spinal segments. All experimental procedures related to the use of macaques were approved by the University of Washington IACUC committee protocol (#4187–07), in accordance with NIH guidelines. Tissue necropsy was performed under the University of Washington IACUC protocol (#4277–01).

An 11 year and 4 months old female *Macaca nemestrina* (10.55 kg) designated for the tissue distribution program was anesthetized with isoflurane after an initial sedation with ketamine. The monkey was paralyzed with a neuromuscular blocker and artificially ventilated. The animal was monitored by a trained surgical technician for pulse oximetry, body temperature, ECG, blood pressure, capnography, and inspired oxygen. External thermal support was provided for the duration of the surgery, and an intravenous line for i.v. drug and isotonic fluid administration, a urethral catheter was inserted to

maintain fluid volume and physiological homeostasis. Under aseptic conditions, a partial laminectomy of the C5-C7 vertebrae was performed to expose the left dorsal surface of the cervical enlargement.

Using a stereotaxic manipulator (Kopf Instruments, Tujunga, CA) on a custom frame, targeted microinjections were performed with a Nanoject II (Drummond Scientific, Broomall, PA). A glass pipette with a broken tip (barrel of tip = 50 × m), filled with rAAV2-CAG-tdTomato, was inserted into the spinal cord through a small longitudinal incision of the dura. Using the dorsal root entry points as a guide, 7 injection tracts spanning dorsal-ventral were used to target the lateral funiculus and ventro-lateral part of the gray matter in the C6/C7 spinal segments. Each tract had five injections (138 nL of virus at 23 nL/second) positioned 100 µm apart spanning –4.1 to –3.7 mm from the surface of the cord. The seven tract locations spanned 2.2 mm anterior-posterior and were in two rows (1 mm apart), evenly spread with slight adjustments to avoid hitting the vasculature. A 1-min wait period was used prior to each first injection within the tract, a 2-min wait period after each injection before moving the pipette, and a 5-min wait period before removing the electrode after the final injection for each tract. The injector's efficacy at ejecting virus was confirmed between each injection tract.

After the injections were complete, artificial dura was placed over the durotomy, the musculature and skin were sutured, and an anti-paralytic agent (atropine) was delivered. Post-operative monitoring and care were performed to minimize pain and distress. Thirty days after the original injections, the animal was anesthetized as described above, and then euthanized using a lethal dose of pentobarbital solution. After death, the animal was perfused transcardially with sodium-free oxygenated ice-cold artificial cerebrospinal fluid (NMDG-aCSF in mM): 92 NMDG, 25 glucose, 30 $NaHCO_3$, 20 HEPES, 10 $MgSO_4$, 2.5 KCl, 1.2 $NaH_2PO_4$, 0.5 $CaCl_2$, 3 sodium pyruvate, 2 thiourea, 5 sodium ascorbate. After perfusion, the brain was removed and the right hemisphere trunk and hand wrist subregions of primary motor cortex were dissected and stored in NMDG-saline on ice. A portion (2 $cm^3$) of the motor cortex was further sub-dissected and stored in freshly made 4% paraformaldehyde in 0.1 M phosphate-buffered saline for later processing.

## Delipidation, immunolabeling, gelation, and expansion
Macaque tissue was processed in the same manner as whole mouse brains.

## Human brain tissue collection, delipidation, labeling, gelation, and expansion
Deidentified postmortem adult human brain tissue (61-year-old male, Hispanic, no known history of neuropsychiatric or neurological conditions) was obtained with permission from next-of-kin by the San Diego Medical Examiner's Office. Tissue procurement was reviewed by the Western Institutional Review Board (WIRB) and did not constitute human subject research requiring Institutional Review Board (IRB) review, in accordance with federal regulation 45 CFR 46 and associated guidance. Postmortem tissue collection was performed in accordance with the Uniform Anatomical Gift Act described in Health and Safety Code §§ 7150, et seq., and other applicable state and federal laws and regulations.

Tissue was manually sliced into 1 cm coronal slabs, flash frozen with liquid nitrogen, vacuum sealed, and stored at –80 °C by the Allen Institute Tissue Processing Team. One slab from the occipital pole was drop-fixed in 4% PFA for approximately 12 hr at 4 °C, and tissue regions containing visual cortex were dissected into ~1 cm blocks for histological processing. Individual blocks were then SHIELD-fixed (Lifecanvas Technologies) prior to sectioning at 100 µm on a sliding freezing microtome to further protect protein antigenicity and tissue architecture. Individual free-floating sections were then passively delipidated (LifeCanvas Technologies) for 1 week prior to immunolabeling. Delipidated sections were subsequently immunolabeled for SMI-32, which identifies heavy chain neurofilaments that make up axon scaffolds in long-range projection neurons. Free-floating sections were blocked in NGSTU (5% goat serum, 0.6% Triton X-100, 4 M urea in 1×PBS) overnight, incubated with a primary antibody (rabbit anti-neurofilament 200, Sigma Aldrich N4142) diluted 1:500 in NGSTU + 0.02% sodium azide for 5 days, followed by a secondary antibody incubation (goat anti-rabbit AF488, Thermo Fisher A-11034) diluted 1:100 in NGST (5% goat serum, 0.6% Triton X-100 in 1×PBS) for 4 days. Due to the thinness (100 um) of this tissue compared to an entire mouse brain, the gelling protocol used varied slightly from the whole brain gelling protocol described above. Most notably, the thermal initiator used was ammonium persulfate (APS) instead of VA-044, the tissue was polymerized

at room temperature for 3 days, and the resulting tissue-hydrogel matrix was digested using a 1:50 concentration of proteinase-k in buffer (5% Sodium Dodecyl Sulfate, 5% Triton X-100, 10% 1 M TRIS pH8).

## Sample mounting

The expanded samples were trimmed to produce smooth edges and then were placed in a custom-built anodized imaging chamber. The sample chamber assembly was performed in a large bath of the expansion and imaging solution (0.05×SSC). The smooth edges of the hydrogel were placed against the chamber panels corresponding to the excitation and emission path, and then the chamber was removed from the bath. A warm (55 °C) solution of 2% agarose, in the same 0.05×SSC as used during expansion, was carefully poured into the chamber space behind the hydrogel for structural rigidity during imaging and let cool to room temperature until solid. The chamber was then sealed and placed in 0.05×SSC for equilibration overnight before imaging. The detailed protocol for mounting the expanded hydrogels is available on protocols.io: (*Ouellette et al., 2023*), https://www.protocols.io/view/whole-mouse-brain-delipidation-immunolabeling-and-cmqbu5sn?step=12.

## Code availability statement

Code is available as listed below. Please see the Materials and methods section for additional usage details. Microscope control: https://github.com/AllenNeuralDynamics/exaspim-control (*Allen Institute for Neural Dynamics, 2025b*) Acquire video streaming: https://github.com/acquire-project Hardware files: https://github.com/AllenNeuralDynamics/exa-spim-hardware (*Allen Institute for Neural Dynamics, 2023b*) Optical files: https://github.com/AllenNeuralDynamics/exa-spim-optics (*Allen Institute for Neural Dynamics, 2023a*) Characterization scripts: https://github.com/AllenNeuralDynamics/exa-spim-characterization (copy archived at *Allen Institute for Neural Dynamics, 2023c*) Compression and cloud transfer: https://github.com/AllenNeuralDynamics/aind-data-transfer.

## Acknowledgements

We thank the project management team at the Allen Institute for Neural Dynamics for helping to coordinate this work; the CZI Imaging Science team for support and collaborative work on the Acquire software project; Stephan Preibisch, Tobias Pietzsch, and the Janelia Open Science Software Initiative (https://www.janelia.org/open-science/overview/open-science-software-initiative-ossi) for support with BigStitcher; Nikita Vladimirov for the field curvature quantification methods; Jon Daniels from Applied Scientific Instrumentation for help and support with ASI related hardware; Keith Russell at Vieworks and Eric Jansenn from Euresys for help with the VP-151MX camera; Peter Majer and Sacha Guyer from Bitplane for assistance with the ImarisWriter API; Magnus Greger, Stuart Singer, Jim Sullivan from Schneider-Kreuznach and Joe Corsi from Navitar for assistance with the metrology lenses; and Tim Wang and Boaz Mohar for feedback on the manuscript. In addition, we thank the Allen Institute Animal Care, Transgenic Colony Management, and Lab Animal Services for mouse husbandry, injections, perfusions; the Allen Institute Tissue Processing Team; the Washington National Primate Research Center veterinary and technical staff; the San Diego Medical Examiner's Office. Supported by the Paul G Allen Foundation and NIH R00CA240681 (AG), RF1MH128841 (KS and JC), U19NS123714 (KS and JC), R01NS123959 (ND and BK), U19NS137920 (partial support for JB), U42OD011123, and P51OD010425 for supporting this work.

## Additional information

### Competing interests

Adam Glaser, Jayaram Chandrashekar, Karel Svoboda: A.G., J.C., and K.S. have filed for patent WO2024129575A1 on aspects of the ExA-SPIM system. The other authors declare that no competing interests exist.

## Funding

| Funder | Grant reference number | Author |
|---|---|---|
| National Cancer Institute | R00CA240681 | Adam Glaser |
| National Institute of Mental Health | RF1MH128841 | Jayaram Chandrashekar<br>Karel Svoboda |
| National Institute of Neurological Disorders and Stroke | R01NS123959 | Brian E Kalmbach<br>Nikolai Dembrow |
| Office of the Director | U42OD011123 | Jonathan T Ting<br>Brian E Kalmbach<br>Nikolai Dembrow |
| National Institute of Neurological Disorders and Stroke | U19NS123714 | Jayaram Chandrashekar<br>Karel Svoboda |
| Office of the Director | P51OD010425 | Jonathan T Ting<br>Steven Perlmutter<br>Brian E Kalmbach<br>Nikolai Dembrow |

The funders had no role in study design, data collection and interpretation, or the decision to submit the work for publication.

## Author contributions

Adam Glaser, Jayaram Chandrashekar, Conceptualization, Resources, Data curation, Software, Formal analysis, Supervision, Funding acquisition, Validation, Investigation, Visualization, Methodology, Writing – original draft, Project administration, Writing – review and editing; Sonya Vasquez, Cameron Arshadi, Judith Baka, Gabor Kovacs, Shamishtaa Seshamani, Anna Grim, Resources, Software, Formal analysis, Investigation, Methodology, Writing – original draft, Writing – review and editing; Rajvi Javeri, Naveen Ouellette, Xiaoyun Jiang, Resources, Formal analysis, Investigation, Methodology, Writing – original draft, Writing – review and editing; Micah Woodard, Nathan Clack, Resources, Software, Formal analysis, Investigation, Methodology, Writing – review and editing; Kevin Cao, Software, Formal analysis, Investigation, Methodology, Writing – original draft, Writing – review and editing; Andrew Recknagel, Resources, Software, Formal analysis, Investigation, Methodology; Pooja Balaram, Lindsey Erion Barner, Resources, Methodology, Writing – original draft, Writing – review and editing; Emily Turschak, Resources, Investigation, Methodology, Writing – original draft, Writing – review and editing; Marcus Hooper, Resources, Software, Investigation, Methodology; Alan Liddell, Resources, Software, Formal analysis; John Rohde, Resources, Formal analysis, Methodology; Ayana Hellevik, Resources, Methodology; Kevin Takasaki, Resources; Molly Logsdon, Investigation, Methodology, Writing – original draft, Writing – review and editing; Chris Chronopoulos, Software, Formal analysis, Investigation, Writing – review and editing; Saskia EJ de Vries, Resources, Software, Formal analysis, Funding acquisition, Writing – original draft, Writing – review and editing; Jonathan T Ting, Steven Perlmutter, Resources, Funding acquisition, Writing – original draft, Writing – review and editing; Brian E Kalmbach, Nikolai Dembrow, Bosiljka Tasic, Resources, Funding acquisition, Investigation, Methodology, Writing – original draft, Writing – review and editing; R Clay Reid, Resources, Software, Formal analysis, Funding acquisition, Investigation, Methodology, Writing – original draft, Writing – review and editing; David Feng, Karel Svoboda, Conceptualization, Resources, Software, Formal analysis, Supervision, Funding acquisition, Investigation, Methodology, Writing – original draft, Project administration, Writing – review and editing

## Author ORCIDs

Adam Glaser ⓘ https://orcid.org/0000-0003-3558-8994
Naveen Ouellette ⓘ http://orcid.org/0009-0005-4472-0880
Kevin Cao ⓘ https://orcid.org/0000-0002-0226-7687
Saskia EJ de Vries ⓘ https://orcid.org/0000-0002-3704-3499
Brian E Kalmbach ⓘ https://orcid.org/0000-0003-3136-8097
Bosiljka Tasic ⓘ https://orcid.org/0000-0002-6861-4506
R Clay Reid ⓘ https://orcid.org/0000-0002-8697-6797

## Ethics

All experimental procedures related to the use of mice were approved by the Institutional Animal Care and Use Committee (IACUC) protocol (#2416) of the Allen Institute for Brain Science, in accordance with National Institutes of Health (NIH) guidelines. All experimental procedures related to the use of macaques were approved by the University of Washington IACUC committee protocol (#4187-07), in accordance with NIH guidelines.

Tissue necropsy was performed under the University of Washington IACUC protocol (#4277-01). De-identified postmortem adult human brain tissue (61 year old male, Hispanic, no known history of neuropsychiatric or neurological conditions) was obtained with permission from next-of-kin by the San Diego Medical Examiner's Office. Tissue procurement was reviewed by the Western Institutional Review Board (WIRB) and did not constitute human subject research requiring Institutional Review Board (IRB) review, in accordance with federal regulation 45 CFR 46 and associated guidance. Postmortem tissue collection was performed in accordance with the Uniform Anatomical Gift Act described in Health and Safety Code section 7150, et seq., and other applicable state and federal laws and regulations.

Reviewer #1 (Public Review): https://doi.org/10.7554/eLife.91979.4.sa1
Reviewer #2 (Public Review): https://doi.org/10.7554/eLife.91979.4.sa2
Author response https://doi.org/10.7554/eLife.91979.4.sa3

## Additional files

### Supplementary files

MDAR checklist

Supplementary file 1. Summary of available electronics metrology technologies.

Supplementary file 2. Data for *Appendix 1—tables 1–3*.

### Data availability

Imaging datasets for this paper are available at:s3://aind-open-data/exaSPIM_708373_2024-04-02_19-49-38    s3://aind-open-data/exaSPIM_671477_2024-01-08_17-03-04    s3://aind-open-data/exaSPIM_Z13288-QN22-26-036_2023-03-26_09-27-21    s3://aind-open-data/exaSPIM_H17.24.006-CX55-31B_2023-05-11_14-59-09 Instructions for how the data is organized and how it can be accessed is available at: https://allenneuraldynamics.github.io/data.html.

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

## Appendix 1

### A Comparisons of theoretical camera sensitivity and speed

Unlike existing SPIM systems, the ExA-SPIM system uses the large-format SONY IMX411 CMOS sensor with different specifications from sCMOS cameras. With respect to sensitivity, the read noise is the most important specification. While the read noise is ~1 *e-* for a sCMOS camera, it is 6.89, 4.57, and 3.48 *e-* for 12, 14, and 16-bit readout modes for the SONY IMX411 sensor. The signal-to-noise ratio (SNR) of a camera sensor can be calculated as:

$$SNR = \frac{QE \times S}{\sqrt{QE \times (S + B) + R^2}} \tag{2}$$

where, QE is the quantum efficiency of the sensor, *S* is the signal in photons, *B* is the background signal in photons, and *R* is the readout noise in *e-*. Plots for an sCMOS and the Sony IMX411 sensor are shown in *Figure 2—figure supplement 3*. These plots yield insights into, for an equivalent number of photons, how the SNR of the sensors would differ, or how many photons each sensor would require to achieve the same SNR. In general, the sCMOS provides much greater SNR at low-light levels (i.e. <50 photons), with diminishing benefits for higher photon levels, especially when compared to the 14- and 16-bit modes of the SONY IMX411 sensor. However, it is also important to consider the SNR as a function of the imaging speed (i.e. pixels/s) of the sensor (*Figure 2e*). For example, when operated with 14-bit readout, the line time of the Sony IMX411 sensor is 20.15 µs. With 14192 pixels per row on the sensor, this corresponds to an imaging speed of $703 \times 10^6$ voxels/s. This contrasts with a traditional sCMOS sensor that only contains 2048 pixels per row, where even at the fastest 4.89 µs line time, the imaging speed is only $418 \times 10^6$ voxels/s. In other words, the Sony IMX411 sensor provides more pixel parallelization within each row. This enables twice the voxel rate with four times the pixel dwell time (i.e. four times the collected signal). It is important to note that sCMOS sensors provide a direct adjustment of the sensor line time, to precisely trade off between sensitivity and imaging speed. For the Sony IMX411 sensor, the line time can be indirectly adjusted by setting the maximum data bandwidth of the camera's data stream. For example, for 16-bit data over entire 14192×10,640 frames, the line time could be set to 100 µs by setting the bandwidth to 284 MB/sec.

**Appendix 1—table 1.** Summary of custom lenses for fluorescence microscopy.

| Reference | Name | Liquid immersion | f | NA | Field of view | Etendue | Working distance |
|---|---|---|---|---|---|---|---|
| *Tang et al., 2024* | Curved LSFM | 1.33–1.56 | 118 | 0.25 | 13.00 | 8.30 | 20.00 |
| Current study | ExA-SPIM | 1.33–1.56 | 100 | 0.31 | 16.40 | 19.65 | 40.00 |
| *Fan et al., 2019* | RUSH | - | - | 0.35 | 14.00 | 18.86 | 19.00 |
| *Ota et al., 2020* | FASHIO-2PM | - | 35 | 0.40 | 4.24 | 2.26 | 4.50 |
| *McConnell et al., 2016* | Mesolens | 1.33–1.56 | 50 | 0.47 | 6.00 | 1.13 | 3.00 |
| *Yu et al., 2024* | Cousa | - | 20 | 0.50 | 2.00 | 0.79 | 20.00 |
| *Yu et al., 2021* | Diesel2P | - | 30 | 0.54 | 6.00 | 8.24 | 8.00 |
| Kyocera | CS06-10-40-154 | 1.52–1.56 | 18 | 0.60 | 2.00 | 1.13 | 40.60 |
| *Sofroniew et al., 2016* | 2p-RAM | - | 21 | 0.60 | 5.00 | 7.07 | 2.70 |
| *Voigt et al., 2024* | Schmidt | 1.33–1.56 | - | 1.00 | 1.15 | 1.04 | 11.00 |

Units
*(f - mm)*
(field of view - mm)
(etendue - mm$^2$)
(working distance - mm)

**Appendix 1—table 2.** Comparison of large-scale volumetric imaging modalities.

| Reference | Imaging method | Optical resolution | Voxel size | Isotropy | Focal volume | Voxel rate |
|---|---|---|---|---|---|---|
| *Economo et al., 2016* | 2 p tomography | 0.45×0.45 × 1.33 | 0.30×0.30 × 1.00 | 2.96:1 | 0.27 | 16 |
| *Gong et al., 2016* | fMOST | 0.32×0.32 × 2.00 | 0.32×0.32 × 2.00 | 6.25:1 | 0.20 | 4 |
| *Narasimhan et al., 2017* | Oblique light-sheet | 0.75×0.75 × 6.90 | 0.41×0.41 × 0.41 | 9.20:1 | 3.88 | 419 |
| *Migliori et al., 2018* | Light-sheet theta | 0.34×0.34 × 3.00 | 0.23×0.23 × 5.00 | 8.82:1 | 0.29 | 105 |
| *Chakraborty et al., 2019* | Axially-swept SPIM | 0.48×0.48 × 0.48 | 0.16×0.16 × 0.16 | 1:1 | 0.11 | 42 |
| *Voleti et al., 2019* | SCAPE 2.0 | 0.60×1.21 × 1.55 | 0.24×0.24 × 0.24 | 2.58:1 | 1.13 | 419 |
| *Gao et al., 2019* | ExLLSM | 0.06×0.06 × 0.09 | 0.03×0.03 × 0.04 | 1.5:1 | <<0.01 | 68 |
| *Chen et al., 2020a* | Multifocal 2 p tomography | 0.36×0.36 × 2.59 | 0.40×0.40 × 1.00 | 7.19:1 | 0.34 | 77 |
| *Guo et al., 2020* | Dual-inverted SPIM | 0.48×0.48 × 0.48 | 0.24×0.24 × 0.24 | 1:1 | 0.11 | 419 |
| *Chen et al., 2020b* | Tiling SPIM | 0.30×0.30 × 2.50 | 0.30×0.30 × 1.50 | 5:1 | 0.22 | 105 |
| *Wang et al., 2021* | fMOST | 0.34×0.34 × 2.00 | 0.23×0.23 × 1.00 | 5.88:1 | 0.23 | 136 |
| *Zhang et al., 2021* | Axially-swept SPIM | 0.95×0.95 × 2.10 | 0.52×0.52 × 1.00 | 2.21:1 | 1.90 | 84 |
| *Xu et al., 2021* | VISOR2 SPIM | 1.00×1.00 × 2.50 | 1.00×1.00 × 2.50 | 2.5:1 | 2.50 | 373 |
| *Glaser et al., 2022* | Open-top light-sheet | 0.45×0.45 × 2.91 | 0.20×0.20 × 0.20 | 6.47:1 | 0.59 | 105 |
| *Qi et al., 2023* | Confocal airy light-sheet | 0.76×0.76 × 2.20 | 0.26×0.26 × 1.06 | 2.08:1 | 1.27 | 419 |
| *Vladimirov et al., 2024* | Benchtop mesoSPIM | 1.50×1.50 × 3.30 | 0.75×0.75 × 2.00 | 2.2:1 | 7.43 | 37 |
| *Tang et al., 2024* | Curved LSFM | 1.22×1.22 × 2.50 | 0.63×0.63 × 1.25 | 2:1 | 3.72 | 130 |
| Current study | ExA-SPIM (1×) | 1.00×1.00 × 3.00 | 0.75×0.75 × 1.00 | 3:1 | 6.75 | 725 |
| Current study | ExA-SPIM (2×) | 0.50×0.50 × 1.50 | 0.38×0.38 × 0.50 | 3:1 | 0.84 | 725 |
| Current study | ExA-SPIM (3×) | 0.33×0.33 × 1.00 | 0.25×0.25 × 0.33 | 3:1 | 0.25 | 725 |
| Current study | ExA-SPIM (4×) | 0.25×0.25 × 0.75 | 0.19×0.19 × 0.25 | 3:1 | 0.11 | 725 |

Units
(optical resolution - μm)
(voxel size - μm)
(focal volume - μm$^3$)
(voxel rate - megavoxels/s)
Note, only modalities capable of providing ≤1.5 μm lateral optical resolution are included.

**Appendix 1—table 3.** ExA-SPIM imaging across scales.

| Manufacturer | Model | M | NA | Lateral resolution | Field of view | Etendue | Speed |
|---|---|---|---|---|---|---|---|
| Schneider-Kreuznach | VEO_JM DIAMOND 1.43×/F3.0 | 1.43 | 0.10 | 3.05 | 57.34 | 26.34 | 2137 |
| Schneider-Kreuznach | VEO_JM DIAMOND 1.67×/F3.0 | 1.67 | 0.10 | 3.05 | 49.10 | 19.32 | 1340 |
| Schneider-Kreuznach | VEO_JM DIAMOND 2.5×/F2.6 | 2.50 | 0.13 | 2.35 | 32.80 | 15.17 | 400 |
| Schneider-Kreuznach | VEO_JM DIAMOND 3.33×/F2.1 | 3.33 | 0.18 | 1.69 | 24.62 | 15.43 | 170 |
| Schneider-Kreuznach | VEO_JM DIAMOND 5.0×/F1.3 | 5.00 | 0.31 | 1.00 | 16.40 | 19.65 | 50 |
| Nikon | Rayfact 1.4 S | 1.40 | 0.11 | 2.77 | 61.71 | 36.19 | 2277 |
| Nikon | Rayfact 1.7 S | 1.70 | 0.11 | 2.77 | 49.37 | 23.16 | 1272 |

*Appendix 1—table 3 Continued on next page*

Appendix 1—table 3 Continued

| Manufacturer | Model | M | NA | Lateral resolution | Field of view | Etendue | Speed |
|---|---|---|---|---|---|---|---|
| Nikon | Rayfact 2.5 S | 2.50 | 0.14 | 2.18 | 34.56 | 18.39 | 400 |
| Nikon | Rayfact 3.5 S | 3.50 | 0.16 | 1.91 | 24.69 | 12.25 | 146 |
| Nikon | Rayfact 5 S | 5.00 | 0.17 | 1.79 | 17.28 | 6.78 | 50 |
| Nikon | Rayfact 1–2 x Variable Lens | 1.00 | 0.09 | 3.39 | 86.4 | 47.49 | 6249 |
| Nikon | Rayfact 1–2 x Variable Lens | 2.00 | 0.12 | 2.54 | 43.2 | 21.11 | 781 |
| Nikon | Rayfact 2–5 x Variable Lens | 2.00 | 0.13 | 2.35 | 43.2 | 24.77 | 781 |
| Nikon | Rayfact 2–5 x Variable Lens | 5.00 | 0.17 | 1.79 | 16.6 | 6.25 | 50 |

Units
(lateral resolution - μm)
(field of view - mm)
(etendue - mm$^2$)
(speed - cm$^3$ per day)

