## [Editor Report · eLife Assessment]

The ExA-SPIM methodology developed here and characterized and supported by **convincing** evidence is an **important** development for the field of light sheet microscopy as the new technology provides an impressive field of view making it possible to image the entire expanded mouse brain at cellular and subcellular resolution.

---

## [Referee Report · Reviewer #1 (Public Review)]

Summary:

Glaser et al present ExA-SPIM, a light-sheet microscope platform with large volumetric coverage (Field of view 85mm^2, working distance 35mm), designed to image expanded mouse brains in their entirety. The authors also present an expansion method optimized for whole mouse brains, and an acquisition software suite. The microscope is employed in imaging an expanded mouse brain, the macaque motor cortex and human brain slices of white matter.

This is impressive work, and represents a leap over existing light-sheet microscopes. As an example, it offers a ~ fivefold higher resolution than mesoSPIM (https://mesospim.org/), a popular platform for imaging large cleared samples. Thus while this work is rooted in optical engineering, it manifests a huge step forward and has the potential to become an important tool in the neurosciences.

Strengths:

-ExA-SPIM features an exceptional combination of field of view, working distance, resolution and throughput.

-An expanded mouse brain can be acquired with only 15 tiles, lowering the burden on computational stitching. That the brain does not need to be mechanically sectioned is also seen as an important capability.

-The image data is compelling, and tracing of neurons has been performed. This demonstrates the potential of the microscope platform.

Review of the revised manuscript:

The authors have carefully addressed my previous concerns and suggestions.

---

## [Referee Report · Reviewer #2 (Public Review)]

In this manuscript, Glaser et al. describe a new selective plane illumination microscope designed to image a large field of view that is optimized for expanded and cleared tissue samples. For the most part, the microscope design follows a standard formula that is common among many systems (e.g. Keller PJ et al Science 2008, Pitrone PG et al. Nature Methods 2013, Dean KM et al. Biophys J 2015, and Voigt FF et al. Nature Methods 2019). The primary conceptual and technical novelty is to use a detection objective from the metrology industry that has a large field of view and a large area camera. The authors characterize the system resolution, field curvature, and chromatic focal shift by measuring fluorescent beads in a hydrogel and then show example images of expanded samples from mouse, macaque, and human brain tissue.

Glaser et al. have responded to the reviewer comments by removing some of the overstated claims from the prior manuscript and editing portions of the manuscript text to enhance the clarity. Although the manuscript would be stronger if the authors had been able to provide data that justified the original high-impact claims from the initial publication (e.g. that the images could be used for robust and automated neuronal tracing across large volumes), the amended manuscript text now more closely matches the supporting data. As with the initial submission, I believe that the microscope design and characterization is a useful contribution to the field and the data are quite stunning.

---

## [Author Response]

The following is the authors’ response to the previous reviews

**Public Reviews:**

**Reviewer #1 (Public Review):**
Summary:Glaser et al present ExA-SPIM, a light-sheet microscope platform with large volumetric coverage (Field of view 85mm^2, working distance 35mm), designed to image expanded mouse brains in their entirety. The authors also present an expansion method optimized for whole mouse brains and an acquisition software suite. The microscope is employed in imaging an expanded mouse brain, the macaque motor cortex, and human brain slices of white matter.This is impressive work and represents a leap over existing light-sheet microscopes. As an example, it offers a fivefold higher resolution than mesoSPIM (https://mesospim.org/), a popular platform for imaging large cleared samples. Thus while this work is rooted in optical engineering, it manifests a huge step forward and has the potential to become an important tool in the neurosciences.Strengths:- ExA-SPIM features an exceptional combination of field of view, working distance, resolution, and throughput.- An expanded mouse brain can be acquired with only 15 tiles, lowering the burden on computational stitching. That the brain does not need to be mechanically sectioned is also seen as an important capability.- The image data is compelling, and tracing of neurons has been performed. This demonstrates the potential of the microscope platform.Weaknesses:- There is a general question about the scaling laws of lenses, and expansion microscopy, which in my opinion remained unanswered: In the context of whole brain imaging, a larger expansion factor requires a microscope system with larger volumetric coverage, which in turn will have lower resolution (Figure 1B). So what is optimal? Could one alternatively image a cleared (non-expanded) brain with a high-resolution ASLM system (Chakraborty, Tonmoy, Nature Methods 2019, potentially upgraded with custom objectives) and get a similar effective resolution as the authors get with expansion? This is not meant to diminish the achievement, but it was unclear if the gains in resolution from the expansion factor are traded off by the scaling laws of current optical systems.

Paraphrasing the reviewer: Expanding the tissue requires imaging larger volumes and allows lower optical resolution. What has been gained?

The answer to the reviewer’s question is nuanced and contains four parts.

First, optical engineering requirements are more forgiving for lenses with lower resolution. Lower resolution lenses can have much larger fields of view (in real terms: the number of resolvable elements, proportional to ‘etendue’) and much longer working distances. In other words, it is currently more feasible to engineer lower resolution lenses with larger volumetric coverage, even when accounting for the expansion factor.

Second, these lenses are also much better corrected compared to higher resolution (NA) lenses. They have a flat field of view, negligible pincushion distortions, and constant resolution across the field of view. We are not aware of comparable performance for high NA objectives, even when correcting for expansion.

Third, although clearing and expansion render tissues ‘transparent’, there still exist refractive index inhomogeneities which deteriorate image quality, especially at larger imaging depths. These effects are more severe for higher optical resolutions (NA), because the rays entering the objective at higher angles have longer paths in the tissue and will see more aberrations. For lower NA systems, such as ExaSPIM, the differences in paths between the extreme and axial rays are relatively small and image formation is less sensitive to aberrations.

Fourth, aberrations are proportional to the index of refraction inhomogeneities (dn/dx). Since the index of refraction is roughly proportional to density, scattering and aberration of light decreases as M^3, where M is the expansion factor. In contrast, the imaging path length through the tissue only increases as M. This produces a huge win for imaging larger samples with lower resolutions.

To our knowledge there are no convincing demonstrations in the literature of diffraction-limited ASLM imaging at a depth of 1 cm in cleared mouse brain tissue, which would be equivalent to the ExA-SPIM imaging results presented in this manuscript.

In the discussion of the revised manuscript we discuss these factors in more depth.

- It was unclear if 300 nm lateral and 800 nm axial resolution is enough for many questions in neuroscience. Segmenting spines, distinguishing pre- and postsynaptic densities, or tracing densely labeled neurons might be challenging. A discussion about the necessary resolution levels in neuroscience would be appreciated.

We have previously shown good results in tracing the thinnest (100 nm thick) axons over cm scales with 1.5 um axial resolution. It is the contrast (SNR) that matters, and the ExaSPIM contrast exceeds the block-face 2-photon contrast, not to mention imaging speed (> 10x).

Indeed, for some questions, like distinguishing fluorescence in pre- and postsynaptic structures, higher resolutions will be required (0.2 um isotropic; Rah et al Frontiers Neurosci, 2013). This could be achieved with higher expansion factors.

This is not within the intended scope of the current manuscript. As mentioned in the discussion section, we are working towards ExA-SPIM-based concepts to achieve better resolution through the design and fabrication of a customized imaging lens that maintains a high volumetric coverage with increased numerical aperture.

- Would it be possible to characterize the aberrations that might be still present after whole brain expansion? One approach could be to image small fluorescent nanospheres behind the expanded brain and recover the pupil function via phase retrieval. But even full width half maximum (FWHM) measurements of the nanospheres' images would give some idea of the magnitude of the aberrations.

We now included a supplementary figure highlighting images of small axon segments within distal regions of the brain.

**Reviewer #2 (Public Review):**
Summary:In this manuscript, Glaser et al. describe a new selective plane illumination microscope designed to image a large field of view that is optimized for expanded and cleared tissue samples. For the most part, the microscope design follows a standard formula that is common among many systems (e.g. Keller PJ et al Science 2008, Pitrone PG et al. Nature Methods 2013, Dean KM et al. Biophys J 2015, and Voigt FF et al. Nature Methods 2019). The primary conceptual and technical novelty is to use a detection objective from the metrology industry that has a large field of view and a large area camera. The authors characterize the system resolution, field curvature, and chromatic focal shift by measuring fluorescent beads in a hydrogel and then show example images of expanded samples from mouse, macaque, and human brain tissue.Strengths:I commend the authors for making all of the documentation, models, and acquisition software openly accessible and believe that this will help assist others who would like to replicate the instrument. I anticipate that the protocols for imaging large expanded tissues (such as an entire mouse brain) will also be useful to the community.Weaknesses:The characterization of the instrument needs to be improved to validate the claims. If the manuscript claims that the instrument allows for robust automated neuronal tracing, then this should be included in the data.

The reviewer raises a valid concern. Our assertion that the resolution and contrast is sufficient for robust automated neuronal tracing is overstated based on the data in the paper. We are hard at work on automated tracing of datasets from the ExA-SPIM microscope. We have demonstrated full reconstruction of axonal arbors encompassing >20 cm of axonal length. But including these methods and results is out of the scope of the current manuscript.

The claims of robust automated neuronal tracing have been appropriately modified.

**Recommendations for the authors:**

**Reviewer #1 (Recommendations For The Authors):**
Smaller questions to the authors:- Would a multi-directional illumination and detection architecture help? Was there a particular reason the authors did not go that route?

Despite the clarity of the expanded tissue, and the lower numerical aperture of the ExA-SPIM microscope, image quality still degrades slightly towards the distal regions of the brain relative to both the excitation and detection objective. Therefore, multi-directional illumination and detection would be advantageous. Since the initial submission of the manuscript, we have undertaken re-designing the optics and mechanics of the system. This includes provisions for multi-directional illumination and detection. However, this new design is beyond the scope of this manuscript. We now mention this in L254-255 of the Discussion section.

- Why did the authors not use the same objective for illumination and detection, which would allow isotropic resolution in ASLM?

The current implementation of ASLM requires an infinity corrected objective (i.e. conjugating the axial sweeping mechanism to the back focal plane). This is not possible due to the finite conjugate design of the ExA-SPIM detection lens.

More fundamentally, pushing the excitation NA higher would result in a shorter light sheet Rayleigh length, which would require a smaller detection slit (shorter exposure time, lower signal to noise ratio). For our purposes an excitation NA of 0.1 is an excellent compromise between axial resolution, signal to noise ratio, and imaging speed.

For other potentially brighter biological structures, it may be possible to design a custom infinity corrected objective that enables ASLM with NA > 0.1.

- Have the authors made any attempt to characterize distortions of the brain tissue that can occur due to expansion?

We have not systematically characterized the distortions of the brain tissue pre and post expansion. Imaged mouse brain volumes are registered to the Allen CCF regardless of whether or not the tissue was expanded. It is beyond the scope of this manuscript to include these results and processing methods, but we have confirmed that the ExA-SPIM mouse brain volumes contain only modest deformation that is easily accounted for during registration to the Allen CCF.

- The authors state that a custom lens with NA 0.5-0.6 lens can be designed, featuring similar specifications. Is there a practical design? Wouldn't such a lens be more prone to Field curvature?

This custom lens has already been designed and is currently being fabricated. The lens maintains a similar space bandwidth product as the current lens (increased numerical aperture but over a proportionally smaller field of view). Over the designed field of view, field curvature is <1 µm. However, including additional discussion or results of this customized lens is beyond the scope of this manuscript.

**Reviewer #2 (Recommendations For The Authors):**
System characterization:- Please state what wavelength was used for the resolution measurements in Figure 2.

An excitation wavelength of 561 nm was used. This has been added to the manuscript text.

- The manuscript highlights that a key advance for the microscope is the ability to image over a very large 13 mm diameter field of view. Can the authors clarify why they chose to characterize resolution over an 8diameter mm field rather than the full area?

The 13 mm diameter field of view refers to the diagonal of the 10.6 x 8.0 mm field of view. The results presented in Figure 1c are with respect to the horizontal x direction and vertical y direction. A note indicating that the 13 mm is with respect to the diagonal of the rectangular imaging field has been added to the manuscript text. The results were presented in this way to present the axial and lateral resolution as a function of y (the axial sweeping direction).

- The resolution estimates seem lower than I would expect for a 0.30 NA lens (which should be closer to ~850 nm for 515 nm emission). Could the authors clarify the discrepancy? Is this predicted by the Zemax model and due to using the lens in immersion media, related to sampling size on the camera, or something else? It would be helpful if the authors could overlay the expected diffraction-limited performance together with the plots in Figure 2C.

As mentioned previously, the resolution measurements were performed with 561 nm excitation and an emission bandpass of ~573 – 616 nm (595 nm average). Based on this we would expect the full width half maximum resolution to be ~975 nm. The resolution is in fact limited by sampling on the camera. The 3.76 µm pixel size, combined with the 5.0X magnification results in a sampling of 752 nm. Based on the Nyquist the resolution is limited to ~1.5 µm. We have added clarifying statements to the text.

- I'm confused about the characterization of light sheet thickness and how it relates to the measured detection field curvature. The authors state that they "deliver a light sheet with NA = 0.10 which has a width of 12.5 mm (FWHM)." If we estimate that light fills the 0.10 NA, it should have a beam waist (2wo) of ~3 microns (assuming Gaussian beam approximations). Although field curvature is described as "minimal" in the text, it is still ~10-15 microns at the edge of the field for the emission bands for GFP and RFP proteins. Given that this is 5X larger than the light sheet thickness, how do the authors deal with this?

The generated light sheet is flat, with a thickness of ~ 3 µm. This flat light sheet will be captured in focus over the depth of focus of the detection objective. The stated field curvature is within 2.5X the depth of focus of the detection lens, which is equivalent to the “Plan” specification of standard microscope objectives.

- In Figure 2E, it would be helpful if the authors could list the exposure times as well as the total voxels/second for the two-camera comparison. It's also worth noting that the Sony chip used in the VP151MX camera was released last year whereas the Orca Flash V3 chosen for comparison is over a decade old now. I'm confused as to why the authors chose this camera for comparison when they appear to have a more recent Orca BT-Fusion that they show in a picture in the supplement (indicated as Figure S2 in the text, but I believe this is a typo and should be Figure S3).

This is a useful addition, and we have added exposure times to the plot. We have also added a note that the Orca Flash V3 is an older generation sCMOS camera and that newer variants exist. Including the Orca BT-Fusion. The BT-Fusion has a read noise of 1.0 e- rms versus 1.6 e- rms, and a peak quantum efficiency of ~95% vs. 85%. Based on the discussion in Supplementary Note S1, we do not expect that these differences in specifications would dramatically change the data presented in the plot. In addition, the typo in Figure S2 has been corrected to Figure S3.

- In Table S1, the authors note that they only compare their work to prior modalities that are capable of providing <= 1 micron resolution. I'm a bit confused by this choice given that Figure 2 seems to show the resolution of ExA-SPIM as ~1.5 microns at 4 mm off center (1/2 their stated radial field of view). It also excludes a comparison with the mesoSPIM project which at least to me seems to be the most relevant prior to this manuscript. This system is designed for imaging large cleared tissues like the ones shown here. While the original publication in 2019 had a substantially lower lateral resolution, a newer variant, Nikita et al bioRxiv (which is cited in general terms in this manuscript, but not explicitly discussed) also provides 1.5-micron lateral resolution over a comparable field of view.

We have updated the table to include the benchtop mesoSPIM from Nikita et al., Nature Communications, 2024. Based on this published version of the manuscript, the lateral resolution is 1.5 µm and axial resolution is 3.3 µm. Assuming the Iris 15 camera sensor, with the stated 2.5 fps, the volumetric rate (megavoxels/sec) is 37.41.

- The authors state that, "We systematically evaluated dehydration agents, including methanol, ethanol, and tetrahydrofuran (THF), followed by delipidation with commonly used protocols on 1 mm thick brain slices. Slices were expanded and examined for clarity under a macroscope." It would be useful to include some data from this evaluation in the manuscript to make it clear how the authors arrived at their final protocol.

Additional details on the expansion protocol may be included in another manuscript.

General comments:There is a tendency in the manuscript to use negative qualitative terms when describing prior work and positive qualitative terms when describing the work here. Examples include:"Throughput is limited in part by cumbersome and error-prone microscopy methods". While I agree that performing single neuron reconstructions at a large scale is a difficult challenge, the terms cumbersome and error-prone are qualitative and lacking objective metrics.

We have revised this statement to be more precise, stating that throughput is limited in part by the speed and image quality of existing microscopy methods.

- The resolution of the system is described in several places as "near-isotropic" whereas prior methods were described as "highly anisotropic". I agree that the ~1:3 lateral to axial ratio here is more isotropic than the 1:6 ratio of the other cited publications. However, I'm not sure I'd consider 3-fold worse axial resolution than lateral to be considered "near" isotropic.

We agree that the term near-isotropic is ambiguous. We have modified the text accordingly, removing the term near-isotropic and where appropriate stating that the resolution is more isotropic than that of other cited publications.

- In the manuscript, the authors describe the photobleaching in their imaging conditions as "negligible". Figure S5 seems to show a loss of 60% fluorescence after 2000 exposures (which in the caption is described as "modest"). I'd suggest removing these qualitative terms and just stating the values.

We agree and have changed the text accordingly.

- The results section for Figure 5 is titled "Tracing axons in human neocortex and white matter". Although this section states "larger axons (>1 um) are well separated... allowing for robust automated and manual tracing" there is no data for any tracing in the manuscript. Although I agree that the images are visually impressive, I'm not sure that this claim is backed by data.

We have now removed the text in this section referring to automated and manual tracing.